# Towards Safe Reinforcement Learning with a Safety Editor Policy

**Haonan Yu, Wei Xu, and Haichao Zhang**
Horizon Robotics
Cupertino, CA 95014
{haonan.yu,wei.xu,haichao.zhang}@horizon.ai

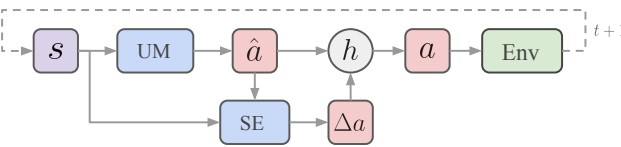

Figure 1: SEditor's framework with two policies utility maximizer (UM) and safety editor (SE). Given a state $s$, UM proposes a preliminary action $\hat{a}$ that aims at maximizing the utility reward. Then SE outputs an adjustment $\Delta a$ to it based on $s$ and $\hat{a}$ itself, to ensure a low constraint violation rate. The action proposal $\hat{a}$ and edit $\Delta a$ are fed to an action editing function $h$ that outputs a final action $a$ to interact with the environment. Both UM and SE are stochastic policies modeled as neural networks and are trained via SGD.

## Abstract

We consider the safe reinforcement learning (RL) problem of maximizing utility with extremely low constraint violation rates. Assuming no prior knowledge or pre-training of the environment safety model given a task, an agent has to learn, via exploration, which states and actions are safe. A popular approach in this line of research is to combine a model-free RL algorithm with the Lagrangian method to adjust the weight of the constraint reward relative to the utility reward dynamically. It relies on a single policy to handle the conflict between utility and constraint rewards, which is often challenging. We present SEditor, a two-policy approach that learns a safety editor policy transforming potentially unsafe actions proposed by a utility maximizer policy into safe ones. The safety editor is trained to maximize the constraint reward while minimizing a hinge loss of the utility state-action values before and after an action is edited. SEditor extends existing safety layer designs that assume simplified safety models, to general safe RL scenarios where the safety model can in theory be arbitrarily complex. As a first-order method, it is easy to implement and efficient for both inference and training. On 12 Safety Gym tasks and 2 safe racing tasks, SEditor obtains much a higher overall safety-weighted-utility (SWU) score than the baselines, and demonstrates outstanding utility performance with constraint violation rates as low as once per 2k time steps, even in obstacle-dense environments. On some tasks, this low violation rate is up to 200 times lower than that of an unconstrained RL method with similar utility performance. Code is available at https://github.com/hnyu/seditor.

36th Conference on Neural Information Processing Systems (NeurIPS 2022).

# 1 Introduction

Safety has been one of the major roadblocks in the way of deploying reinforcement learning (RL) to the real world. Although RL for strategy and video games has achieved great successes (Silver et al., 2016; Vinyals et al., 2019; Berner et al., 2019), the cost of executing actions that lead to catastrophic failures in these cases is low: at most losing a game. On the other hand, RL for control has also mostly been studied in virtual simulators (Brockman et al., 2016; Tassa et al., 2020). Aside from the data collection consideration, to circumvent the safety issue (*e.g.*, damages to real robots and environments) is another important cause of using these simulators. When RL is sometimes applied to real robots (Kim et al., 2004; Levine et al., 2016; OpenAI et al., 2019), data collection and training are configured in restrictive settings to ensure safety. Thus to promote the deployment of RL to more real-world scenarios, safety is a critical topic. In this paper, we study a safe RL problem of maximizing utility with extremely low constraint violation rates.

There are two general settings of safe RL. Some existing works make the assumption that the environment safety model is a known prior. The agent is able to query an oracle function to see if any state is safe or not, *without* actually visiting that state. This includes having access to a well calibrated dynamics model (Berkenkamp et al., 2017; Chow et al., 2018; Thomas et al., 2021) or the set of safe/unsafe states (Turchetta et al., 2020; Luo and Ma, 2021; Li et al., 2021). Or they assume that a pre-training stage of safe/unsafe states or policies is performed on offline data, and then the learned safety knowledge is transferred to main tasks (Dalal et al., 2018; Miret et al., 2020; Thananjeyan et al., 2021). With these assumptions, they are able to achieve few or even zero constraint violations during online exploration. However, such assumptions also put restrictions on their applicable scenarios.

This paper focuses on another setting where no prior knowledge or pre-training of safety models is assumed. The agent only gets feedback on which states are unsafe from its exploration experience. In other words, it has to learn the safety model (implicitly or explicitly) from scratch and the constraint budget can only be satisfied *asymptotically*. Although we cannot completely avoid constraint violations during online exploration, this setting can be especially helpful for sim-to-real transfer (Zhao et al., 2020), or when the training process can be protected from damages. Prior works combine model-free RL (Bhatnagar and Lakshmanan, 2012; Ray et al., 2019; Tessler et al., 2019; Bohez et al., 2019; Stooke et al., 2020; Zhang et al., 2020; Qin et al., 2021) or model-based RL (As et al., 2022) with the Lagrangian method (Bertsekas, 1999). This primal-dual optimization dynamically adjusts the weight of the constraint reward relative to the utility reward, depending on how well the violation rate target is being met. A single policy is then trained from a weighted combination of the utility and constraint rewards. However, reconciling utility maximization with constraint violation minimization usually poses great challenges to this single policy.

In this paper, we present **SEditor**, a general safe RL approach that decomposes policy learning across two polices (Figure 1). The utility maximizer (UM) policy is only responsible for maximizing the utility reward without considering the constraints. Its output actions are potentially unsafe. The safety editor (SE) policy then transforms these actions into safe ones. It is trained to maximize the constraint reward while minimizing a hinge loss of the utility state-action values before and after an action is edited. Both UM and SE are trained in an off-policy manner for good sample efficiency. Our two-policy paradigm is largely inspired by existing safety layer designs (Dalal et al., 2018; Pham et al., 2018; Cheng et al., 2019; Li et al., 2021) which simplify (*e.g.*, to linear or quadratic) environment safety models. The high-level idea is the same though: modifying a utility-maximizing action only when necessary.

We evaluate SEditor on 12 Safety Gym (Ray et al., 2019) tasks and 2 safe car racing tasks adapted from Brockman et al. (2016), targeting at *very* low violation rates. SEditor obtains a much higher overall safety-weighted-utility (SWU) score (defined in Section 4) than four baselines. It demonstrates outstanding utility performance with constraint violation rates as low as once per 2k time steps, even in obstacle-dense environments. Our results reveal that the two-policy cooperation is critical, while simply doubling the size of a single policy network will not lead to comparable results. The choices of the action distance function and editing function are also important in certain circumstances. In summary, our contributions are:

a) We extend the existing safety layer works to more general safe RL scenarios where the environment safety model can in theory be arbitrarily complex.

b) We present SEditor, a first-order, easy-to-implement approach that is trained by SGD like most model-free RL methods. It is efficient during both inference and training, as it does not solve a multi-step inner-level optimization problem when transforming an unsafe action into a safe one.

c) When measuring the distance between an action and its edited version, we show that in some cases the hinge loss of their state-action values is better than the usual L2 distance (Dalal et al., 2018; Pham et al., 2018; Li et al., 2021) in the action space. We further show that an additive action editing function introduces an effective inductive bias for the safety editor.

d) We propose the safety-weighted-utility (SWU) score for quantitatively evaluating a safe RL method. The score is a soft indicator of the *dominance* defined by Ray et al. (2019).

e) Finally, we achieve outstanding utility performance even with an extremely low constraint violation rate ($5 \times 10^{-4}$) in dense-obstacle environments. On some tasks, this low violation rate is up to 200 times lower than that of a unconstrained RL method with similar utility performance.

## 2 Preliminaries

The safe RL problem can be defined as policy search in a constrained Markov decision process (CMDP) $(\mathcal{S}, \mathcal{A}, p, r, c)$. The state space $\mathcal{S}$ and action space $\mathcal{A}$ are both assumed to be continuous. The environment transition function $p(s_{t+1}|s_t, a_t)$ determines the probability density of reaching $s_{t+1} \in \mathcal{S}$ after taking action $a_t \in \mathcal{A}$ at state $s_t \in \mathcal{S}$. The initial state distribution $\mu(s_0)$ determines the probability density of an episode starting at state $s_0$. Both $p(s_{t+1}|s_t, a_t)$ and $\mu(s_0)$ are usually unknown to the agent. For every transition $(s_t, a_t, s_{t+1})$, the environment outputs a scalar $r(s_t, a_t, s_{t+1})$ which we call the *utility* reward. Sometimes one uses the expected reward of taking $a_t$ at $s_t$ as $r(s_t, a_t) \triangleq \mathbb{E}_{s_{t+1} \sim p(\cdot|s_t, a_t)} r(s_t, a_t, s_{t+1})$ for a simpler notation. Similarly, the environment also outputs a scalar $c(s_t, a_t)$ as the cost. To unify the reward and cost notations, we define the *constraint* reward $r_c(s_t, a_t) \triangleq -c(s_t, a_t) \leq 0$, and the CMDP becomes $(\mathcal{S}, \mathcal{A}, p, r, r_c)$. For both utility and constraint rewards, the value is *the higher the better*. Finally, we denote the agent's policy as $\pi(a_t|s_t)$ which dictates the probability density of taking $a_t$ at $s_t$.

For each $s_t$, the utility state value of following $\pi$ is denoted by $V^\pi(s_t) = \mathbb{E}_{\pi, p} \sum_{t'=t}^{\infty} \gamma^{t'-t} r(s_{t'}, a_{t'})$, and the utility state-action value is denoted by $Q^\pi(s_t, a_t) = r(s_t, a_t) + \gamma \mathbb{E}_{s_{t+1} \sim p} V^\pi(s_{t+1})$, where $\gamma \in [0, 1)$ is the discount for future rewards. Similarly, we can define $V_c^\pi$ and $Q_c^\pi$ for the constraint reward. Then we consider the safe RL objective:

$$\max_\pi \mathbb{E}_{s_0 \sim \mu} V^\pi(s_0), \quad s.t. \quad \mathbb{E}_{s_0 \sim \mu} V_c^\pi(s_0) + C \geq 0, \tag{1}$$

where $C \geq 0$ is the constraint violation budget. It might be unintuitive to specify $C$ for a discounted return, so one can rewrite

$$\mathbb{E}_{\mu, \pi, p} \sum_{t=0}^{\infty} \gamma^t (r_c(s_t, a_t) + c) \geq 0, \tag{2}$$

and specify the per-step budget $c$ instead, relating to $C$ by $\sum_{t=0}^{\infty} \gamma^t c = \frac{c}{1-\gamma} = C$. Note that $c$ is not strictly imposed on every step. Instead, it is only in the average sense (by a discount factor), treated as the *violation rate target*.

The Lagrangian method (Bertsekas, 1999) converts the constrained optimization problem Eq. 1 into an unconstrained one by introducing a multiplier $\lambda$:

$$\min_{\lambda \geq 0} \max_\pi \left[ \mathbb{E}_{s_0 \sim \mu} V^\pi(s_0) + \lambda \left( \mathbb{E}_{s_0 \sim \mu} V_c^\pi(s_0) + C \right) \right]. \tag{3}$$

Intuitively, it dynamically adjusts the weight $\lambda$ according to how well the constraint state value satisfies the budget, by evaluating (approximately) the difference

$$\Lambda_\pi \triangleq \mathbb{E}_{s_0 \sim \mu} V_c^\pi(s_0) + C = \mathbb{E}_{\mu, \pi, p} \sum_{t=0}^{\infty} \gamma^t (r_c(s_t, a_t) + c), \tag{4}$$

which is the gradient of $\lambda$ given $\pi$ in Eq. 3. When optimizing $\pi$ given $\lambda$, Eq. 3 becomes

$$\max_\pi \mathbb{E}_{s_0 \sim \mu} [V^\pi(s_0) + \lambda V_c^\pi(s_0)] = \max_\pi \mathbb{E}_{\mu, \pi, p} \sum_{t=0}^{\infty} \gamma^t (r(s_t, a_t) + \lambda r_c(s_t, a_t)).$$

Thus $\lambda$ can be seen as the weight of the constraint reward when it is combined with the utility reward to convert multi-objective RL into single-objective RL. This objective can be solved by typical model-free RL algorithms. For practical implementations, previous works (Ray et al., 2019; Tessler et al., 2019; Bohez et al., 2019) usually perform gradient ascent on the parameters of $\pi$ and gradient descent on $\lambda$ simultaneously, potentially with different learning rates.

## 3 Approach

We consider a pair of cooperative policies. The first policy utility maximizer (UM) denoted by $\pi_\phi$, optimizes the utility reward by proposing a preliminary action $\hat{a} \sim \pi_\phi(\cdot|s)$ which is potentially unsafe. The second policy safety editor (SE) denoted by $\pi_\psi$, edits the preliminary action by $\Delta a \sim \pi_\psi(\cdot|s, \hat{a})$ to ensure safety, and the result action $a = h(\hat{a}, \Delta a)$ is output to the environment, where $h$ represents an editing function. Note that we condition SE on UM's output $\hat{a}$. Together the two policies cooperate to maximize the agent's utility while maintaining a safe condition (Figure 1). For simplicity, we will denote the overall composed policy by $\pi_{\psi \circ \phi}(a|s)$.

**Motivation.** SEditor decomposes a difficult policy learning task that maximizes both utility and safety, into two easier subtasks that focus on either utility or safety, based on the following considerations:

a) Different effective horizons. In most scenarios, safety requires either responsive actions (Dalal et al., 2018), or planning a number of steps ahead to prevent the agent entering non-recoverable states. Thus SE's actual decision horizon could be short depending on the nature of the safety constraints. This is in contrast to UM's decision horizon which is usually long for goal achieving. In other words, we could expect the optimization problem of SE to be easier than that of UM, and SE has a chance of being learned faster if separated from UM.

b) Guarded exploration. From the perspective of UM, its MDP (precisely, action space) is altered by SE. UM's actions are guarded by the barriers set up by the SE. Instead of UM being discouraged for an unsafe action (*i.e.*, punished with negative signal), SE gives suggestions to UM by redirecting the unsafe action to a safe but also utility-high action to continue its exploration. This guarded exploration leads to a better overall exploration strategy because safety constraints are less likely to hinder UM's exploration (Figure 4 illustration).

**Objectives.** We employ an off-policy actor-critic setting for training the two policies. Given an overall policy $\pi_{\psi \circ \phi}$, we can use typical TD backup to learn $Q^{\pi_{\psi \circ \phi}}(s, a)$ and $Q_c^{\pi_{\psi \circ \phi}}(s, a)$ parameterized as $Q(s, a; \theta)$ and $Q_c(s, a; \theta)$ respectively, where we use $\theta$ to collectively represent the network parameters of the two state-action values. Given $s_{t+1} \sim p(\cdot|s_t, a_t)$ and $a_{t+1} \sim \pi_{\psi \circ \phi}(\cdot|s_{t+1})$, the Bellman backup operator (point estimate) for the utility state-action value is

$$\mathcal{T}^{\pi_{\psi \circ \phi}} Q(s_t, a_t; \theta) = r(s_t, a_t) + \gamma Q(s_{t+1}, a_{t+1}; \theta), \tag{5}$$

with the backup operator for the constraint state-action value $Q_c$ defined similarly with $r_c$. Both $Q(s, a; \theta)$ and $Q_c(s, a, ; \theta)$ can be learned on transitions $(s_t, a_t, s_{t+1})$ sampled from a replay buffer.

For off-policy training of $\phi$ and $\psi$, we first transform Eq. 3 into a bi-level optimization surrogate as:

$$\text{(a)} \quad \max_{\phi, \psi} \left[ \mathbb{E}_{s \sim \mathcal{D}, a \sim \pi_{\psi \circ \phi}(\cdot|s)} (Q(s, a; \theta) + \lambda Q_c(s, a; \theta)) \right], \quad \text{(b)} \quad \min_{\lambda \geq 0} \lambda \Lambda_{\pi_{\psi \circ \phi}}. \tag{6}$$

$\mathcal{D}$ denotes a replay buffer and $\Lambda_{\pi_{\psi \circ \phi}}$ is defined by Eq. 4. We basically have a historical marginal state distribution for training the policies, but still *use the initial state distribution $\mu$ for training $\lambda$*. The motivation for this difference is, when tuning $\lambda$, we should always care about how well the policy satisfies our constraint budget starting with $\mu$ but not with some historical state distribution.

We continue transforming the off-policy objective (Eq. 6, a) into two,

$$\text{(a)} \quad \max_{\phi} \mathbb{E}_{\substack{s \sim \mathcal{D}, \hat{a} \sim \pi_\phi(\cdot|s), \\ \Delta a \sim \pi_\psi(\cdot|s, \hat{a}), \\ a = h(\hat{a}, \Delta a)}} \left[ Q(s, a; \theta) \right], \quad \text{(b)} \quad \max_{\psi} \mathbb{E}_{\substack{s \sim \mathcal{D}, \hat{a} \sim \pi_\phi(\cdot|s), \\ \Delta a \sim \pi_\psi(\cdot|s, \hat{a}), \\ a = h(\hat{a}, \Delta a)}} \left[ -d(a, \hat{a}) + \lambda Q_c(s, a; \theta) \right], \tag{7}$$

where $d(a, \hat{a})$ is a distance function measuring the change from $\hat{a}$ to $a$. It is not necessarily proportional to $\Delta a$ because the editing function $h$ could be nonlinear. The role of SE $\pi_\psi$ is to maximize the constraint reward while minimizing some distance between the actions before and after the modification. The role of UM $\pi_\phi$ is to only maximize the utility reward. However, it only can only do

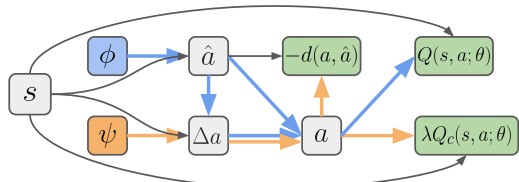

Figure 2: The computational graph of Eq. 7. Nodes denote variables and edges denote operations. The green blocks are negative losses, the blue paths are the (reversed) gradient paths of $\phi$, and the orange paths are the (reversed) gradient paths of $\psi$. Paths in black or blue are detached for $\psi$, and paths in black or orange are detached for $\phi$.

so *through the lens of* SE $\pi_\psi$. In other words, SE has actually changed the action space (also MDP) of UM. This property corresponds to the motivation of guarded exploration mentioned earlier.

We would like to emphasize that, unlike the safety layer (Dalal et al., 2018) which projects unsafe actions based on instantaneous safety costs, the training objective (Eq. 7, b) for SE relies on a safety critic $Q_c$ which is learned as the expected future constraint return. Therefore, maximizing this critic will take into account long-term safety behaviors. In other words, if there are non-recoverable states where SE can't do anything to ensure safety, it will edit the agent's actions long before that to avoid entering those states.

**Action editing function.** We choose the editing function $h$ to be mainly additive and non-parametric. Without loss of generality, we assume a bounded action space $[-A, A]^M$, and that both $\hat{a}$ and $\Delta a$ are already in this space. Then we define $a = h(\hat{a}, \Delta a) = \min(\max(\hat{a} + 2\Delta a, -A), A)$, where $\min$ and $\max$ are element-wise. The multiplication by 2 and the clipping make sure that $a \in [-A, A]^M$, namely, SE has a full control over the final action in that it can overwrite UM's action if necessary. Because SE can be arbitrarily complex, its output $\Delta a$ could depend on the current state $s$ and the action proposal $\hat{a}$ in an arbitrarily complex way. Thus even though the additive operation is simple, the overall editing process is already general enough to represent any modification.

This additive editing function is motivated by *constraint sparsity*. Usually, constraint violations are only triggered for some states. Most often, the action proposal by UM is already safe if the agent is far away from obstacles. To explicitly introduce this inductive bias, we use the additive editing function which ensures that the majority of SE's modifications are close to 0. This makes the optimization landscape of SE easier (see Figure 12 Appendix F for some empirical observations).

**Distance function.** Prior works on safety layer design, such as Dalal et al. (2018); Pham et al. (2018); Li et al. (2021), set the distance function $d(\cdot, \cdot)$ as the L2 distance. Later we will show that L2 is not always the best option. Instead, we use the hinge loss of the utility state-action values of $\hat{a}$ and $a$:

$$d(a, \hat{a}) \triangleq \max(0, Q(s, \hat{a}; \theta) - Q(s, a; \theta)) \tag{8}$$

This loss is zero if the edited action $a$ already obtains a higher utility state-action value than the preliminary action $\hat{a}$. In this case, only the constraint Q is optimized by $\pi_\psi$. Otherwise, the inner part of Eq. 7 (b) is recovered as $Q(s, a; \theta) + \lambda Q_c(s, a; \theta)$, as we can drop the term $-Q(s, \hat{a}; \theta)$ due to its gradient *w.r.t.* $\psi$ being zero. Our distance function in the utility Q space is more appropriate than the L2 distance in the action space, because eventually we care about how the utility changes after the action is edited. The L2 distance between the two actions is only an approximation to the change, from the perspective of the Taylor series of $Q(s, a; \theta)$.

**Evaluating $\Lambda_{\pi_{\psi \circ \phi}}$.** Given a batch of rollout experiences $\{(s_n, a_n)\}_{n=1}^N$ following $\pi_{\psi \circ \phi}$, we approximate the gradient of $\lambda$ (Eq. 4) as

$$\Lambda_{\pi_{\psi \circ \phi}} \approx \frac{1}{N} \sum_{n=1}^N r_c(s_n, a_n) + c, \tag{9}$$

where $c$ is the violation rate target defined in Eq. 2. Namely, after every rollout, we collect a batch of constraint rewards, compare each of them to $-c$, and use the mean of differences to adjust $\lambda$. This approximation allows us to update $\lambda$ using mini-batches of data instead of having to wait for whole episodes to finish, or having to rely on estimated $V_c^{\pi_{\psi \circ \phi}}$ which is usually not accurate. To reduce temporal correlation (which might affect the constraint evaluation) in the rollout batch data, we use multiple parallel environments (Appendix H). This rollout batch is also the one that will be put into the replay buffer.

**Training.** To practically train the objectives, we apply SGD to Eq. 6 (b) and Eq. 7 simultaneously, resulting in a first-order method for approximated bi-level optimization (Likhosherstov et al., 2021). SGD is made possible by applying the re-parameterization trick (Appendix G) to both $\hat{a} \sim \pi_\phi$ and $\Delta a \sim \pi_\psi$. A computational graph of Eq. 7 is illustrated in Figure 2. We would like to highlight that a big difference between our SE and the previous safety layers (Dalal et al., 2018; Pham et al., 2018) is that SE directly uses a feedforward prediction $\Delta a \sim \pi_\psi$ (Eq. 7, a) to replace the optimization result of the objective of transforming unsafe actions into safe ones. In contrast, previous safety layers construct closed-form solutions or inner-level differentiable optimization steps (*e.g.*, quadratic programming (Amos and Kolter, 2017)) for simplified safety models at every inference step. Our overall approach generalizes to various constraint rewards (safety models) and action distance functions. Finally, to encourage exploration, we add the entropy terms of $\pi_\phi$ and $\pi_\psi$ into Eq. 7, with their weights dynamically adjusted according to two entropy targets following Haarnoja et al. (2018).

Both UM and SE are trained from scratch in our experiments. Here we discuss an alternative situation where an existing policy is pre-trained to maximize utility, and we use it as the initialization for UM. In this case, UM *cannot* be frozen because its MDP (and hence its optimum) will be changed by the evolving SE. Instead, it should be fine-tuned to adapt to the changing behaviors of SE. Potentially, a pre-trained UM will speed up the convergence of SEditor. We will leave this to our future work.

## 4 Experiments

**Baselines.** We compare SEditor with four baselines.

- *PPO-Lag* combines PPO (Schulman et al., 2017) with the Lagrangian method, as done in Ray et al. (2019). Since *PPO* is an on-policy algorithm, we expect its sample efficiency to be much lower than off-policy algorithms.
- *FOCOPS* (Zhang et al., 2020) is analogous to PPO-Lag with two differences: 1) there is no clipping of the importance ratio, and 2) a KL divergence regularization term with a fixed weight is added to the policy improvement loss, with an early stopping when this term averaged over a rollout batch data violates the trust region constraint.
- *SAC-actor2x-Lag* combines SAC (Haarnoja et al., 2018) with the Lagrangian method. Similar to SEditor, SAC-actor2x-Lag trains its policy on states sampled from the replay buffer but trains $\lambda$ on states generated by the *rollout policy*. Its gradient of $\lambda$ is also estimated by Eq. 9. The policy network size is doubled. This is to match the model capacity of having two actors in SEditor.
- *SAC* serves as an unconstrained optimization baseline to calibrate the utility return.

We choose not to compare with second-order CMDP approaches such as CPO (Achiam et al., 2017) or PCPO (Yang et al., 2020) because they require (approximately) computing the inverse of the Fisher information matrix, which is prohibitive when the parameter space is large. Especially for our CNN based policies, second-order methods are impractical.

To analyze the key components of SEditor, we also evaluate two variants of it for ablation studies.

- *SEditor-L2* defines $d(a, \hat{a}) \triangleq \|a - \hat{a}\|^2$ as done in most prior works of safety layer.
- *SEditor-overwrite* makes SE directly overwrite UM's action proposal by $a = h(\hat{a}, \Delta a) = \Delta a$.

All compared approaches including the variants of SEditor, share a common training configuration (*e.g.*, replay buffer size, mini-batch size, learning rate, etc) as much as possible. Specific changes are made to accommodate to particular algorithm properties (Appendix H).

**Evaluation metric.** Following Ray et al. (2019), one method *dominates* another if "it strictly improves on either return or cost rate and does at least as well on the other". Accordingly, we emphasize that the utility reward or constraint violation rate should never be compared in isolation, as one could easily find a method that optimizes either metric very well. Particularly in the experiments, we set the learning rate of the Lagrangian multiplier $\lambda$ much larger than that of the remaining parameters. This ensures that any continued constraint violation will lead to a quick increase of $\lambda$ and drive the violation rate back to the target level. This conservative strategy makes some compared methods have similar constraint violation curves, but vastly different utility performance. For a quantitative comparison of their final performance, we compute the *safety-weighted-utility* (SWU) scores of the compared methods towards the end of training. The score is a soft indicator of dominance, and it is a product of two ratios:

$$\text{SWU} \triangleq \min\left\{1, \frac{\text{ConstraintViolationRateTarget}}{\text{ConstraintViolationRate}}\right\} \times \frac{\text{UtilityScore}}{\text{UtilityScore}_{\text{UnconstrainedRL}}}.$$

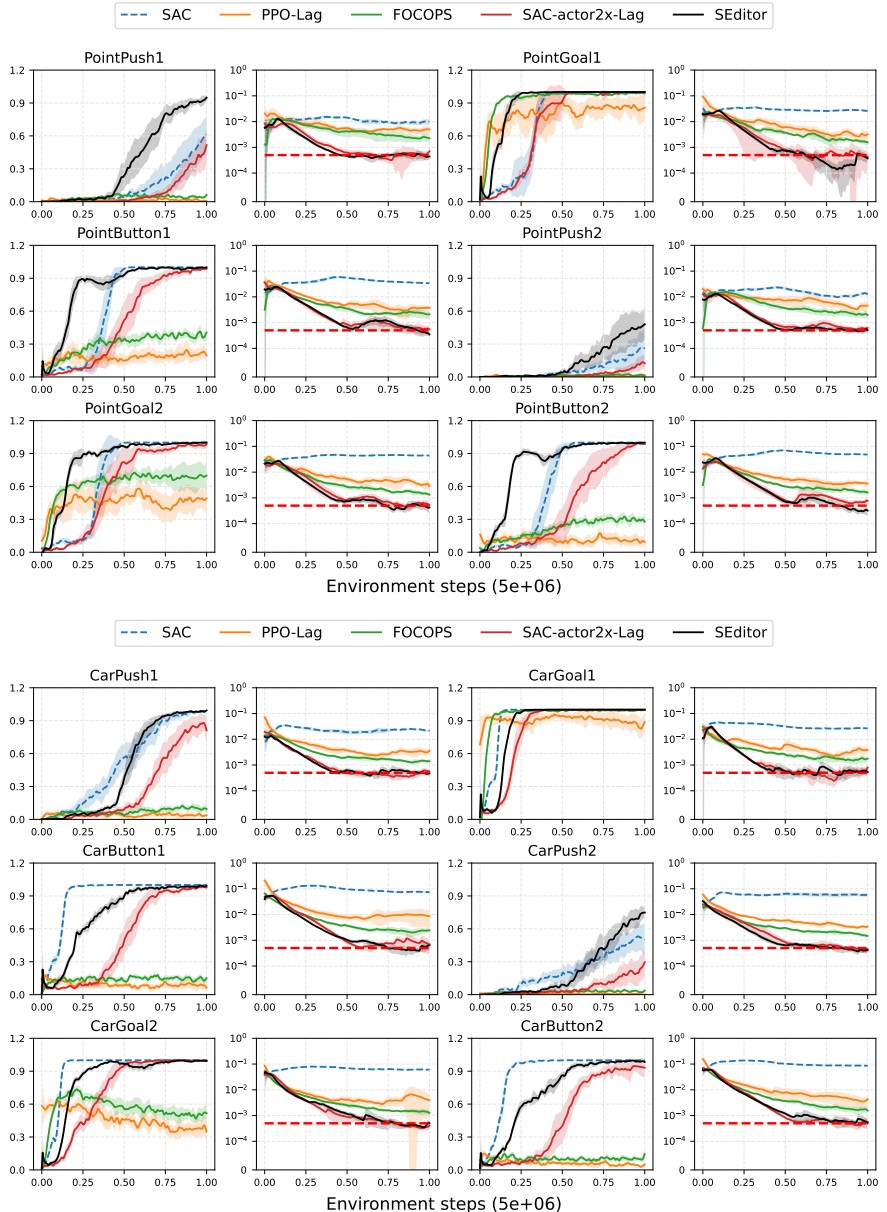

Figure 3: The training curves of the 12 Safety Gym tasks. Odd columns: ↑ success rate. Even columns: ↓ constraint violation rate (log scale). Red dashed horizontal lines: violation rate target $c = 5 \times 10^{-4}$. Shaded areas: 95% confidence interval (CI).

The reason for having $\min\{1, \cdot\}$ is because we only care when the violation rate is higher than the target. In practice, UtilityScore can be the episodic utility return (assuming positive) or the success rate. We choose $\text{UtilityScore}_{\text{SAC}}$ as $\text{UtilityScore}_{\text{UnconstrainedRL}}$. In our experiments, both UtilityScore and ConstraintViolationRate are averaged over the last $\frac{1}{10}$ of the training steps to reduce variances.

**Safety Gym tasks.** In Safety Gym (Ray et al., 2019), a robot with lidar sensors navigates through cluttered environments to achieve goals. We use the POINT and CAR robots in our experiments. Either of them has three tasks: GOAL, BUTTON, and PUSH. Each task has two levels, where level 2 has more obstacles and a larger map size than level 1. In total we have $3 \times 2 \times 2 = 12$ tasks. Whenever an obstacle is in contact with the robot, a constraint reward of $-1$ is given. Thus the constraint violation rate can be calculated as the negative average constraint reward. The utility reward is calculated as the decrement of the distance between the robot (GOAL and BUTTON) or box

| | Safety Gym | | | | | | | | | | | | Safe Racing | | Overall | Improvement |
|---|---|---|---|---|---|---|---|---|---|---|---|---|---|---|---|---|
| | CP1 | CG1 | CB1 | CP2 | CG2 | CB2 | PP1 | PG1 | PB1 | PP2 | PG2 | PB2 | SR | SRO | | |
| SAC | 0.02 | 0.02 | 0.01 | 0.01 | 0.01 | 0.01 | 0.05 | 0.02 | 0.01 | 0.04 | 0.01 | 0.01 | 0.14 | 0.03 | 0.03 | 3567% |
| PPO-Lag | 0.01 | 0.11 | 0.00 | 0.00 | 0.04 | 0.00 | 0.00 | 0.13 | 0.03 | 0.00 | 0.09 | 0.01 | 0.04 | 0.04 | 0.04 | 2650% |
| FOCOPS | 0.03 | 0.28 | 0.03 | 0.03 | 0.20 | 0.05 | 0.02 | 0.32 | 0.10 | 0.01 | 0.26 | 0.09 | 0.04 | 0.04 | 0.11 | 900% |
| SAC-actor2x-Lag | 0.74 | 0.63 | 0.70 | 0.59 | 1.00 | 0.81 | 0.60 | 1.00 | 0.89 | 0.37 | 0.94 | 0.64 | 0.37 | 0.08 | 0.67 | 64% |
| SEditor | **1.01** | **0.94** | 0.78 | **1.49** | 0.99 | **0.95** | **1.55** | 1.00 | 1.00 | **1.78** | 1.00 | **1.00** | **1.28** | **0.57** | **1.10** | - |

Table 1: The SWU scores of different methods. Task name abbreviations: CP - CARPUSH, CG - CARGOAL, CB - CARBUTTON, PP - POINTPUSH, PG - POINTGOAL, PB - POINTBUTTON, SR - SAFERACING, and SRO - SAFERACINGOBSTACLE.

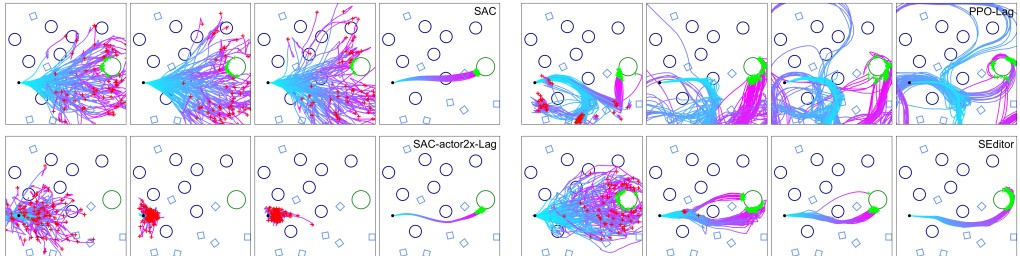

Figure 4: Visualization of rollout trajectories at different training stages. For each method from left to right, the trajectories are generated by checkpoints at 10%, 20%, 30%, and 50% of the training process. On each trajectory, time flows from cyan to purple. A red cross + denotes a failed trajectory while a green cross + denotes a successful trajectory. Each bird's-eye map is a sketch of the environment layout. The small black dot is the robot's initial position. The big green circle is the goal location and the other shapes are obstacles. In summary, SEditor has a better exploration strategy because the barriers set up by SE can redirect UM's unsafe actions to continue its exploration, while SAC-actor2x-Lag's exploration will be hindered in when encountering similar obstacles.

(PUSH) and the goal at every step. We define a success as finishing a task within a time limit of 1000. We emphasize that the agent has *no* prior knowledge of which states are unsafe. The map layout is *randomized* at the beginning of each episode. This randomization poses a great challenge to the agent because it has to generalize the safety knowledge to new scenarios.

We customize the environments to equip the agent with a more advanced lidar sensor in two ways:

i) It is a natural lidar instead of a pseudo one used by Ray et al. (2019). The natural lidar simulates a ray intersecting with an object from an origin. The pseudo lidar can only detect object centers, while the natural lidar reveals shape information of objects, which is required to achieve a very low constraint threshold.

ii) Our lidar has 64 bins while the original lidar has only 16 bins. The 16-bin pseudo lidar's low precision is a bottleneck of achieving much fewer safety violations. Without perception precision, the problem is ill-posed for a robot suffering from blind spots to avoid touching objects at all.

We refer the reader to Appendix A.1 for more details of the environment and tasks. Our customized Safety Gym is available at `https://github.com/hnyu/safety-gym`.

We set the constraint violation target $c = 5 \times 10^{-4}$, meaning that the agent is allowed to violate any constraint only once every 2k steps on average. This threshold is only $\frac{1}{50}$ of the threshold $c = 0.025$ used by the original Safety Gym experiments (Ray et al., 2019), highlighting the difficulty of our task. Each compared approach is trained with 9 random seeds. The training curves are in Figure 3 and the SWU scores are in Table 1. We see that SEditor obtains much higher SWU scores than the baselines on 7 of the 12 tasks, while being comparable on the rest. While SAC-actor2x-Lag performs well among the baselines, its double-size policy network does not lead to results comparable to SEditor. This suggests that SEditor does not simply rely on the large combined capacity of two policy networks to improve the performance, instead, its framework in Figure 1 matters. Both PPO-Lag and FOCOPS missed all the constraint violation rate targets. The unconstrained baseline SAC obtains better success rate sample efficiency than SEditor with the CAR robot, but violates constraints much more. Surprisingly, SAC is worse than SEditor with the POINT robot regarding success rates, even without constraints. Finally, we highlight that towards the end of training on CARBUTTON1/2, CARPUSH2, and CARGOAL2, SEditor violates constraints up to 200 times less than SAC does, while achieving success rates on par with SAC (black curves *vs.* dashed blue curves)!

To analyze the exploration behaviors of SAC, SAC-actor2x-Lag, PPO-Lag, and SEditor, we visualize their rollout trajectories on an example map of POINTGOAL2, at different training stages (the

training is done on randomized maps but here we only evaluate on one map). For each approach, the checkpoints at 10%, 20%, 30%, and 50% of the training process are evaluated and 100 rollout trajectories are generated for each of them. The trajectories are then drawn on a bird's-eye sketch map (Figure 4). We can see that although SAC is able to find the goal location in the very beginning, it tends to explore more widely regardless of the obstacles, and its final paths ignore obstacles. SAC-actor2x-Lag learns to respect the constraints after 10% of training, but the obstacles greatly hinder its exploration: the trajectories are confined in a small region. As a result, it takes some time for it to find the goal location. PPO-Lag is slower at learning the constraints and it even hits obstacles at 50%. SEditor is able to quickly explore regions between obstacles and refine the navigation paths passing them instead of being blocked.

Our ablation studies (Appendix D Figure 8) show that, in this particular safe RL scenario, SEditor is not sensitive to the choice of distance function: SEditor-L2 achieves similar results with SEditor. However, the editing function does make a difference: SEditor-overwrite is clearly worse than SEditor in terms of utility performance. This shows that the inductive bias of the edited action $a$ being close to the preliminary action $\hat{a}$ is very effective (see Appendix F). Our two-stage "propose-and-edit" strategy is more efficient than outputting an action in one shot.

**Safe racing tasks.** Our next two tasks, SAFERACING and SAFERACINGOBSTACLE, are adapted from the unconstrained car racing task in Brockman et al. (2016). The goal of either task is to finish a racetrack as fast as possible. The total reward of finishing a track is 1000, and it is evenly distributed on the track tiles. In SAFERACING, the car has to stay on the track and receives a constraint reward of $-1$ whenever driving outside of it. In SAFERACINGOBSTACLE, the car receives a constraint reward of $-1$ if it hits an obstacle on the track, but there is no penalty for being off-track. An episode finishes after every track tile is visited by the car, or after 1000 time steps. The track (length and shape) and obstacles (positions and shapes) are *randomly* generated for each episode. As in Safety Gym, this randomization could expose a never-experienced safety scenario to the agent at any time. The agent's observations include a bird's-eye view image and a car status vector. Note that this high-dimensional input space usually poses great challenges to second-order safe RL methods. Again, the constraint violation rate can be computed as the negative average constraint reward. We set the constraint violation rate target $c = 5 \times 10^{-4}$. For evaluation, we use undiscounted episode return as the UtilityScore. Each compared approach is trained with 9 random seeds. The training curves are shown in Figure 5 and SWU scores in Table 1. SEditor has much higher SWU scores than all baselines. Moreover, it is the only one that satisfies the harsh violation rate target towards the end of training. Surprisingly, even with constraints SEditor gets a much better utility return than SAC on SAFERACING. One reason is that without the out-of-track penalty as racing guidance, the car easily gets lost on the map and collects lots of meaningless timesteps. Although SAC gets a much higher return on SAFERACINGOBSTACLE, it greatly violates the constraint budget. Interestingly, the ablation studies show results that are complementary to those on the Safety Gym tasks (Appendix D Figure 9). Now the action distance function makes a big difference. Changing it to the L2 distance greatly impacts the utility return, especially on SAFERACINGOBSTACLE where almost no improvement is made. This demonstrates that the closeness of two actions does not necessarily reflect the closeness of their state-action values (Appendix E).

## 5 Related Work

Safe RL is closely related to multi-objective RL (Roijers et al., 2013), where the agent optimizes a scalarization of multiple rewards given a preference (Van Moffaert et al., 2013), or finds a set of policies covering the Pareto front if no preference is provided (Moffaert and Nowé, 2014). In our case, the preference for the constraint reward is always changing, because we try to maintain it to a certain level instead of maximizing it.

Trust-region methods (Achiam et al., 2017; Chow et al., 2019; Yang et al., 2020; Liu et al., 2022) for solving CMDPs put a constraint on the KL divergence between the new and old policies, where the KL divergence is second-order approximated. For solving the surrogate objective at each iteration, the inverse of the Fisher information matrix is approximated. Liu et al. (2022) decomposes the policy update step of CPO (Achiam et al., 2017) into two steps, where the first E-step computes a non-parametric form of the optimal policy and the second M-step distills it to a parametric policy. These methods often involve complex formulations and high computational costs. In contrast, SEditor is a conceptually simple and computationally efficient first-order method.

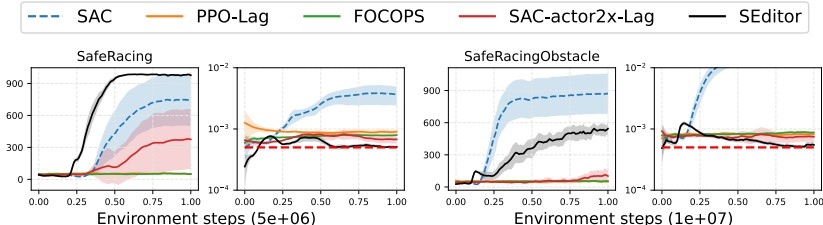

Figure 5: The training curves of the safe racing tasks. Odd columns: $\uparrow$ undiscounted episode return. Even columns: $\downarrow$ constraint violate rate (log scale). Red dashed horizontal lines: violation rate target $c = 5 \times 10^{-4}$. Shaded areas: 95% confidence interval (CI).

The line of works closest to ours is safety layer (Dalal et al., 2018; Pham et al., 2018; Cheng et al., 2019; Li et al., 2021). Although they also transform unsafe actions into safe ones, their safety layers assume simple safety models which can be solved in closed forms or by inner-level optimization (*e.g.*, quadratic programming (Amos and Kolter, 2017)). We extend this idea to more general safe RL scenarios without assuming a particular safety model. Our SE can be also treated as a teacher under the teacher-student framework, where UM is the student whose actions are corrected by SE. However, existing works (Turchetta et al., 2020; Langlois and Everitt, 2021) under this framework more or less rely on various heuristics and task-specific rules, or have access to the simulator's internal state. In complex realistic environments, such assumptions are invalidated.

Recovery RL (Thananjeyan et al., 2021) also decomposes policy learning across two policies. It makes a *hard* switch between a task policy and a recovery policy by comparing a pre-trained constraint critic with a manual threshold. It requires an offline dataset of demonstrations to learn the constraint critic and fix it during online exploration. This offline training requirement and the hard switch scheme reduce the flexibility of Recovery RL. DESTA (Mguni et al., 2021) fully decouples utility from safety by introducing two separate agents. The safety agent can decide at which states it takes control of the system while the task agent can only maximize utility at the remaining states. There is hardly any communication or cooperation between the two agents. In contrast, SEditor lets UM and SE cooperate with a "propose-and-edit" strategy, instead of switching between them. Flet-Berliac and Basu (2022) also adopts a two-policy design. However, their two policies are adversarial. More specifically, one of their policies tries to intentionally maximize risks by behaving unsafely, to shrink the feasibility region of the agent's value function. Their overall policy will bias towards being conservative. Our SE minimizes risks, and UM and SE always cooperate to resolve the conflict between utility and safety.

# 6 Limitations and Conclusions

While SEditor is able to maintain an extremely low constraint violation rate in expectation, oscillation does happen occasionally on the training curves around the target threshold (Figure 3). In general, if no oracle safety model can be accessed by the agent, avoiding this oscillation issue would be very difficult. On one hand, our tasks always generate new random environment layouts for new episodes, and in theory the agent has to generalize its safety knowledge (*e.g.*, one obstacle being dangerous at this location usually implies it being also dangerous even when the background changes). On the other hand, neural networks have the notorious issue of catastrophic forgetting, meaning that while the agent learns some new safety knowledge, it might forget that already learned. Finally, from a practical point of view, as the violation target reaches 0, it takes an exponential number of samples to evaluate if the agent truly can meet the target. These challenging issues are currently ignored by SEditor and would be interesting future directions.

In summary, we have introduced SEditor, for general safe RL scenarios, that makes a safety editor policy transform preliminary actions output by a utility maximizer policy into safe actions. In problems where a utility maximizing action is often safe already, this architecture leads to faster policy learning. We train this two-policy framework with simple primal-dual optimization, resulting in an efficient first-order approach. On 14 safe RL tasks with very harsh constraint violation rates, SEditor achieves an outstanding overall SWU score. We hope that SEditor can serve as a preliminary step towards safe RL with harsh constraint budgets.

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
