



POINTGOAL2      POINTBUTTON2      POINTPUSH2



Figure 6: Left: The three level-2 Safety Gym tasks with the POINT robot. Level-1 tasks have less crowded maps. Green zones: goal locations; blue zones: hazard zones; cyan cubes: vases; purple cubes: gremlins; orange cylinders: buttons; blue cylinders: pillars. Hazard zones are penetrable areas. Vases are lightweight and can be moved by the robot. Unlike other static obstacles, gremlins have circular movements. Right: the POINT and CAR robots.

|  | GOAL | PUSH | BUTTON |
|---|---|---|---|
| POINT | 204 | 268 | 268 |
| CAR | 216 | 280 | 280 |

Table 2: The observation dimensions of our custom Safety Gym tasks. For each combination, level 1 and 2 have the same observation space. All action spaces are $[-1, 1]^2$.

## A  Task details

### A.1  Safety Gym

In Safety Gym (Ray et al., 2019) environments, a robot with lidar sensors navigates through cluttered environments to achieve tasks. There are three types of tasks (Figure 6) for a robot:

1) GOAL: reaching a goal location while avoiding hazard zones and vases.
2) BUTTON: hitting one goal button out of several buttons while avoiding gremlins and hazard zones.
3) PUSH: pushing a box to a goal location while avoiding pillars and hazard zones.

Each task has two levels, where level 2 has more obstacles and a larger map size than level 1. In total there are $3 \times 2 = 6$ tasks for a robot. We use the POINT and CAR robots in our experiments.

We customized the environment so that the robot has a natural lidar of 64 bins. The natural lidar contains more information of object shapes in the environment than the default pseudo lidar. We found that rich shape information is necessary for the agent to achieve a harsh constraint threshold. A separate lidar vector of length 64 is produced for each obstacle type or goal. All lidar vectors and the robot status vector (*e.g.*, acceleration, velocity, rotations) are concatenated together to produce a flattened observation vector. A summary of the observation dimensions is in Table 2. Whenever an obstacle is in contact with the robot, a constraint reward of $-1$ is given. The utility reward is calculated as the decrement of the distance between the robot (GOAL and BUTTON) or box (PUSH) and the goal at every step. An episode terminates when the goal is achieved, or after 1000 time steps. We define a success as achieving the goal before timeout. The map layout is *randomized* at the beginning of each episode. We emphasize that the agent has no prior knowledge of which states are unsafe, thus path planning with known obstacles does not apply here. Our customized Safety Gym is available at `https://github.com/hnyu/safety-gym`.

### A.2  Safe Racing

The agent's observation includes a bird's-eye view image ($96 \times 96$) and a car status vector (length 11) consisting of ABS sensor, wheel angles, speed, angular velocity, and the remaining tile portion. The action space is $[-1, 1]^3$. Based on the code `https://github.com/NotAnyMike/gym/blob/master/gym/envs/box2d/car_racing.py`, we modify the original unconstrained car rac-

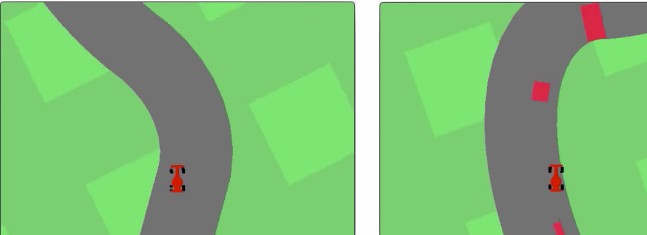

Figure 7: The SAFERACING (left) and SAFERACINGOBSTACLE (right) tasks. In the former task, the car needs to keep itself within the track to avoid penalties. In the latter task, the car gets a collision cost once hitting an obstacle (red blocks). However, it can drive outside of the track without being penalized. For each episode, the track layout and obstacles are randomly generated.

ing task (Brockman et al., 2016) to add obstacles on the track. A bird's-eye view of the two safe racing tasks is illustrated in Figure 7. We set the obstacle density ($\frac{\text{\#Obstacles}}{\text{\#TrackTiles}}$) to $10\%$ for SAFERACIN-GOBSTACLE.

## B    Training with 10M Environment Steps

In Figure 3, one might wonder if SEditor only improves sample efficiency of the Lagrangian SAC, but doesn't really improve the final performance on the 12 Safety Gym tasks. To answer this question, we report here the SWU score comparison between SAC-actor2x-Lag and SEditor with 10M steps on Safety Gym. The SWU scores are calculated based on the performance of unconstrained SAC at 10M steps (and thus are not directly comparable to those in Table 1).

| | CP1 | CG1 | CB1 | CP2 | CG2 | CB2 | PP1 | PG1 | PB1 | PP2 | PG2 | PB2 | Overall | Improvement |
|---|---|---|---|---|---|---|---|---|---|---|---|---|---|---|
| SAC-actor2x-Lag | 0.91 | 0.84 | **1.00** | 0.82 | 0.84 | 0.80 | 1.02 | 1.00 | 0.74 | 0.75 | 1.00 | 1.00 | 0.89 | 17% |
| SEditor | 1.01 | **1.00** | 0.85 | **1.28** | **1.00** | **0.98** | 0.94 | 1.00 | 0.97 | **1.42** | 1.00 | 1.00 | **1.04** | - |

Table 3: The SWU scores of SAC-actor2x-Lag and SEditor at 10M environment steps. Task name abbreviations: CP - CARPUSH, CG - CARGOAL, CB - CARBUTTON, PP - POINTPUSH, PG - POINTGOAL, and PB - POINTBUTTON.

We observe that both methods have saturated at 10M steps. Training more steps somewhat decreases but not closes the gap between SAC-actor2x-Lag and SEditor regarding the final performance.

## C    Experiment on the Unmodified POINTGOAL1

Since we have modified the Safety Gym environments to pursue a much (98%) lower constraint violation threshold, one might be curious to see if SEditor also performs well on the original unmodified tasks. As a representative experiment, we compare SEditor (averaged over 4 random seeds) with the results reported in Ray et al. (2019) and Stooke et al. (2020) on the unmodified POINTGOAL1. We observe that all methods can satisfy the constraint threshold well; the difference resides in their utility performance. We list their (rough) utility scores at different environment steps below:

| Steps | Ray et al. (2019) (PPO-Lag) | Ray et al. (2019)(TRPO-Lag) | Stooke et al. (2020) | SEditor |
|---|---|---|---|---|
| $2.5 \times 10^7$ | - | - | 26 | 29 |
| $1 \times 10^7$ | 13 | 17 | 23 | 27 |
| $5 \times 10^6$ | 14 | 16 | 22 | 24 |

Table 4: The utility performance on the original POINTGOAL1. All methods are able to satisfy the constraint threshold of $0.025$.

It's unsurprising that SEditor did pretty well under such a much higher cost limit. We also observe that without P-control, SEditor's cost curve is similar to the ones of $K_P = 0$ in Stooke et al. (2020), which is expected: the initial cost was high and then quickly dropped to the limit. Our cost stabilized at about 3M steps while Stooke et al. (2020) stabilized at about 10M steps (with $K_I = 1 \times 10^{-2}$).

# D   Ablation Study Results

Figure 8 and 9 show the comparison results between SEditor and its two variants SEditor-L2 and SEditor-overwrite, as introduced in Section 4. All three approaches share a common training setting except the changes to the action distance function $d(a, \hat{a})$ or the editing function $h(\hat{a}, \Delta a)$.

We notice that Figure 8 and 9 have opposite results, where SEditor-L2 is comparable to SEditor in the former while SEditor-overwrite is comparable to SEditor in the latter. This suggests a hypothesis that the magnitude of $\Delta a$ output by SE is usually small in Safety Gym but larger in the safe racing tasks, because SEditor-overwrite removes the inductive bias of $\hat{a}$ being close to $h(\hat{a}, \Delta a)$. To verify this hypothesis, we record the output $\Delta a$ when evaluating the trained models of SEditor on two representative tasks POINTPUSH1 and SAFERACINGOBSTACLE. For either task, we plot the empirical distribution of $\Delta a$ over 100 episodes (each episode has 1000 steps). The plotted distributions are in Figure 10. It is clear that on POINTPUSH1, the population of $\Delta a$ is more centered towards 0. On SAFERACINGOBSTACLE, the population tends to distribute on the two extremes of $\pm 1$. This somewhat explains why the L2 distance can be a good proxy for the utility Q closeness on Safety Gym but not on the safe racing tasks.

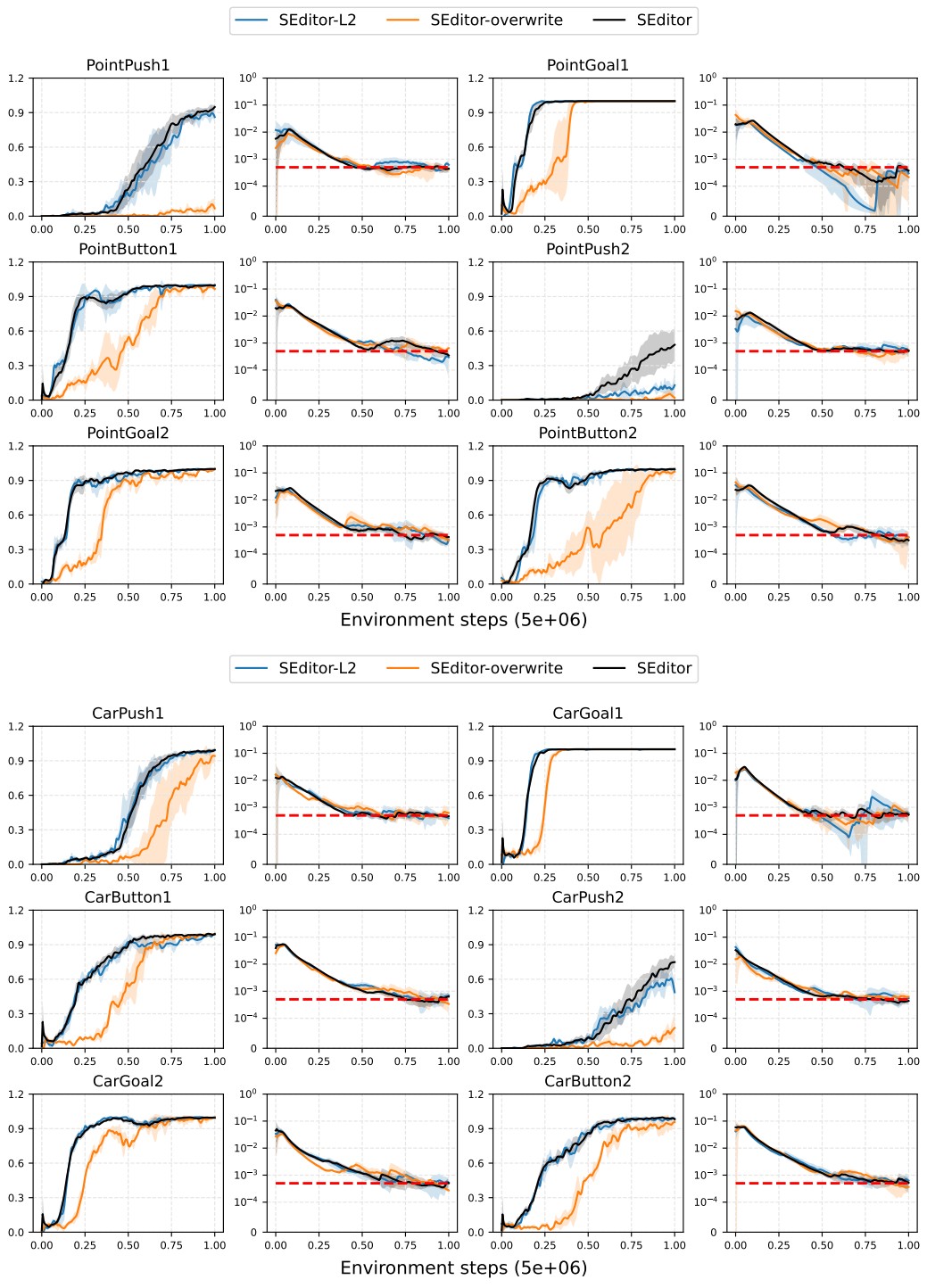

Figure 8: The ablation study results on the 12 Safety Gym tasks. Odd columns: ↑ success rate. Even columns: ↓ constraint violation rate (log scale). Red dashed horizontal lines: violation rate target $c = 5 \times 10^{-4}$. Shaded areas: 95% confidence interval (CI).

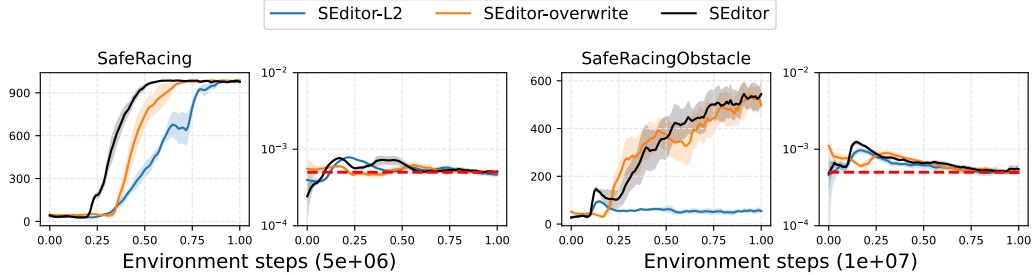

Figure 9: The ablation study results on the safe racing tasks. Odd columns: ↑ undiscounted episode return. Even columns: ↓ constraint violation rate (log scale). Red dashed horizontal lines: violation rate target $c = 5 \times 10^{-4}$. Shaded areas: 95% confidence interval (CI).

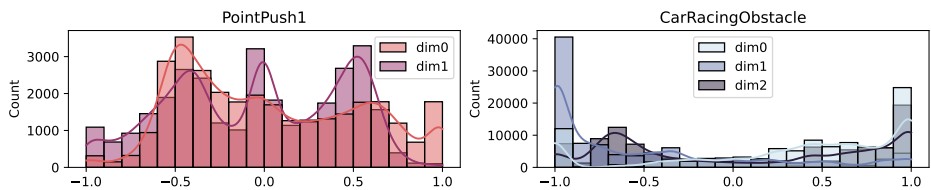

Figure 10: The empirical distributions of $\Delta a$ over 100 episodes of POINTPUSH1 and SAFERACINGOBSTACLE, by evaluating trained models of SEditor. Recall that their action dimensions are 2 and 3, respectively.

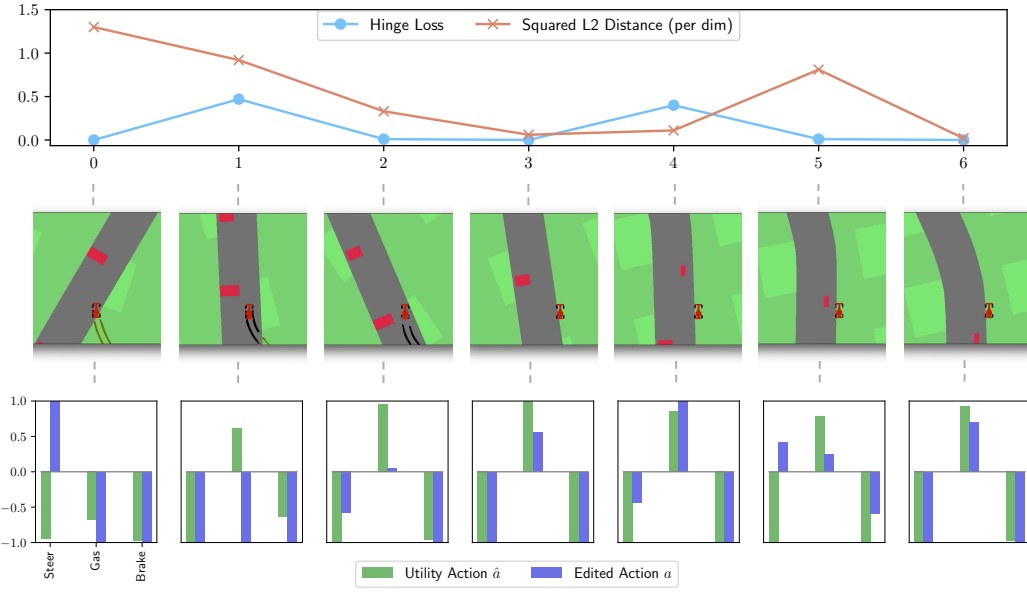

Figure 11: Inference results during an example episode of SAFERACINGOBSTACLE, by evaluating a trained model of SEditor. Top: The changing hinge loss of the utility state-action values of $\hat{a}$ and $a$, and their changing squared L2 distance (per dimension). Note that the absolute magnitudes of the two quantities are not comparable. Instead, only the relative trend within either curve is meaningful. Middle: 7 key frames of the episode (zoom in for a better view). Bottom: bar plots of the utility action $\hat{a}$ by UM and the edited action $a$ by SE. Each plot corresponds to a key frame.

| Action dimension | $-1$ | $+1$ |
|:---:|:---:|:---:|
| 0 | Steer left | Steer right |
| 1 | No acceleration | Full acceleration |
| 2 | No brake | Full brake |

Table 5: The semantics of the action space of SAFERACINGOBSTACLE. Since the action space is continuous, numbers between the two extremes of $\pm 1$ represent a smooth transition.

## E  Hinge Loss *vs.* L2 distance

In Figure 9, SEditor-L2 is especially bad compared to SEditor, indicating that the L2 distance is not a good choice for measuring the difference between the utility action $\hat{a}$ and the edited action $a$, when we actually attempt to compare their state-action values. For further analysis, we evaluate a trained SEditor model on SAFERACINGOBSTACLE, and inspect the following inference results during an episode:

a) The action proposed by UM $\pi_\phi$, also known as the utility action $\hat{a} \in [-1, 1]^3$;
b) The edited action $a \in [-1, 1]^3$ by SE $\pi_\psi$ as the output to the environment;
c) The hinge loss of their utility state-action values (Eq. 8);
d) The squared L2 distance (per dimension) of the two actions $\frac{1}{3}\|\hat{a} - a\|^2$

We select 7 key frames of the episode and visualize their corresponding inference results in Figure 11. The semantics of the action space is listed in Table 5.

In the first frame, the car just gets back on track from outside and there is an obstacle in front of it. The utility action $\hat{a}$ steers left while the edited action $a$ steers right due to safety concern. This causes their squared L2 distance to be quite large. However, $Q(s, a; \theta)$ is no worse than $Q(s, \hat{a}; \theta)$, and thus in this case $\pi_\psi$ of SEditor only needs to focus on maximizing the constraint reward, while $\pi_\psi$ of SEditor-L2 has to make compromises. The second frame is where the hinge loss is positive because $\hat{a}$ commands acceleration while $a$ does not, resulting in a potential decrease of the utility return. (The front tires of the car are already steered all the way to the right, thus both actions turn left.) Overall, SE is more cautious and wants to slow down when passing the obstacle. For frames 2 and 3, $\hat{a}$ and $a$ are similar, as the car is temporarily free from constraint violation. Frame 4 is an example where a subtle difference in the L2 distance results in a large hinge loss. The car is driving near the border of the track, and at any time it could go off-track and miss the next utility reward (a reward is given if the car touches a track tile). Thus $\hat{a}$ turns all the way to the left to make sure that the off-track scenario will not happen. However, because there is an obstacle in front, $a$ makes the steering less extreme. Since the car is at the critical point of being on-track, even a small difference in steering results in a large hinge loss. In comparison, the left-front and left-rear tires are already on the track in frame 5, and even though $a$ wants to turn right a little bit to avoid the obstacle, the utility return is not affected and the hinge loss is still zero. Frame 6 is an example where both the L2 distance and hinge loss are small.

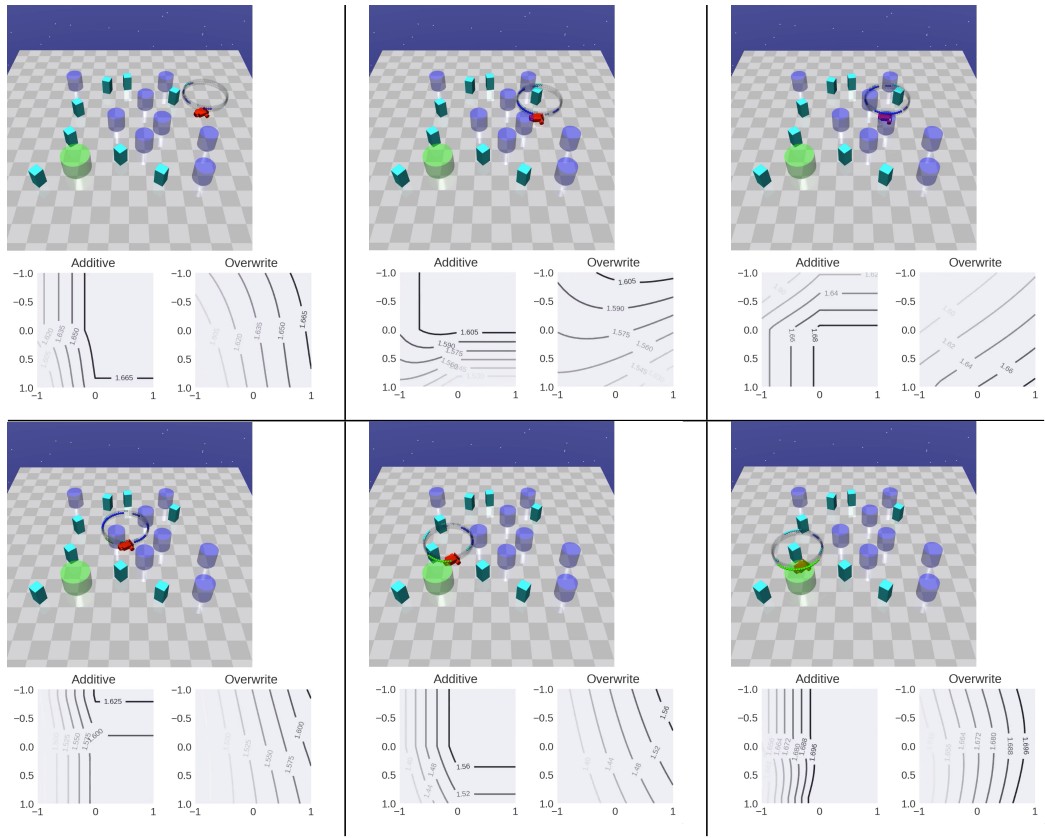

Figure 12: Several key frames of an evaluation episode of SEditor. Each frame is paired with two visualized optimization landscapes, one using an additive action editing function while the other directly overwriting the action proposal $\hat{a}$ with $\Delta a$. Each contour map is generated by enumerating $\Delta a$ given a sampled action proposal $\hat{a}$ from UM $\pi_\phi$, and evaluating the value of of Eq. 7 (b). The map axes correspond to the agent's action dimensions (2D). Note that because we normalize the constraint reward by its moving average mean and standard deviation (Table 6), the values on the map can be positive.

## F    Action Editing Function

In Section 3, our motivation for an additive action editing function is to ensure an easier optimization landscape for SE $\pi_\psi$. We hypothesize that if each proposed $\hat{a}$ is "mostly" safe, then there is an inductive bias for $\pi_\psi$ to output $\Delta a \to 0$. In Section 4 on the Safety Gym tasks, we did observe that SEditor-overwrite (directly using $\Delta a$ as the final action) is worse than SEditor. To further show why an additive editing function is beneficial, when evaluating SEditor we visualize the function surface of Eq. 7 (b) *w.r.t.* $\Delta a$ given a sampled action proposal $\hat{a}$ in Figure 12. It is clear that compared to an overwriting editing function, an additive editing function always has a much larger set of optimal $\Delta a$. Furthermore, this set almost always covers those $\Delta a$ that are close to 0. Thus the additive editing function does provide a very good inductive bias for SE $\pi_\psi$.

## G    Parameterization

We set $\lambda = \text{softplus}(\lambda_0)$ to enforce the Lagrangian multiplier $\lambda \geq 0$, where $\lambda_0$ is a real-valued variable. Thus Eq. 6 (b) becomes

$$\min_{\lambda_0}(\text{softplus}(\lambda_0)\Lambda_{\pi_{\psi \circ \phi}}) \tag{10}$$

as unconstrained optimization solved by typical SGD.

We parameterize both policies $\pi_\phi$ and $\pi_\psi$ as Beta distribution policies (Chou et al., 2017). The advantage of Beta over Normal (Haarnoja et al., 2018) is that it natively has a bounded support of $[0,1]$ for a continuous action space. It avoids using squashing functions like tanh which could have numerical issues when computing the inverse mapping. With PyTorch, we can use the reparameterization trick for the Beta distribution to enable gradient computation in Eq. 7. Generally speaking, Eq. 7 is re-written as

$$\text{(a)} \quad \max_{\phi} \quad \mathbb{E}_{\substack{s \sim \mathcal{D}, \epsilon_1, \epsilon_2, \\ a = h(f_\phi(s,\epsilon_1), f_\psi(s, f_\phi(s,\epsilon_1), \epsilon_2))}} \Big[ Q(s, a; \theta) \Big],$$

$$\text{(b)} \quad \max_{\psi} \quad \mathbb{E}_{\substack{s \sim \mathcal{D}, \epsilon_1, \epsilon_2, \\ a = h(f_\phi(s,\epsilon_1), f_\psi(s, f_\phi(s,\epsilon_1), \epsilon_2))}} \Big[ - d(a, f_\phi(s,\epsilon_1)) + \lambda Q_c(s, a; \theta) \Big],$$

where $\epsilon_1$ and $\epsilon_2$ are sampled from two fixed noise distributions. Then gradients can be easily computed for $\phi$ and $\psi$.

# H Hyperparameters and Compute

In this section, we list the key hyperparameters used by the baselines and SEditor. A summary for the Safety Gym experiments is in Table 6. For the safe racing experiments, we only list the differences with Table 6 in Table 7. For the remaining implementation details, we refer the reader to the source code `https://github.com/hnyu/seditor`.

With the model hyperparameters and training configurations above, a single job (one random seed) of each compared method in Section 4 takes up to 6 hours training on any of the Safety Gym tasks and up to 20 hours training on either safe racing task, on a single machine of Intel(R) Core(TM) i9-7960X CPU@2.80GHz with 32 CPU cores and one RTX 2080Ti GPU. In practice, we use our internal cluster with similar hardware to launch multiple jobs in parallel.

| Hyperparameter | PPO-Lag | FOCOPS | SAC | SAC-actor2x-Lag | SEditor |
|---|---|---|---|---|---|
| Number of parallel environments | 32 | ← | ← | ← | ← |
| Initial rollout steps before training | N/A | N/A | 10000 | ← | ← |
| Number of hidden layers* | 3 | ← | ← | ← | ← |
| Number of hidden units of each layer* | 256 | ← | ← | ← | ← |
| Beta distribution min concentration | 1.0 | ← | ← | ← | ← |
| Frame stacking | 4 | ← | ← | ← | ← |
| Reward normalizer clipping° | 10.0 | ← | ← | ← | ← |
| Hidden activation | tanh | ← | ← | ← | ← |
| Entropy regularization weight | $10^{-3}$ | N/A | N/A | N/A | N/A |
| Entropy target per dimension | N/A | N/A | $-1.609^\dagger$ | ← | $(-1.609, -1.609)$ |
| KLD weight‡ | N/A | 1.5 | N/A | N/A | N/A |
| Trust region bound‡ | N/A | 0.02 | N/A | N/A | N/A |
| Initial Lagrangian multiplier $\lambda$ | 1.0 | ← | ← | ← | ← |
| Learning rate of $\lambda$ | 0.01 | ← | ← | ← | ← |
| Learning rate▷ | $10^{-4}$ | ← | $3 \times 10^{-4}$ | ← | ← |
| Training interval (action steps per environment) | 8 | ← | 5 | ← | ← |
| Mini-batch size | 256 | ← | 1024 | ← | ← |
| Mini-batch length for n-TD or GAE | 8 | ← | ← | ← | ← |
| TD($\lambda$) for n-TD or GAE | 0.95 | ← | ← | ← | ← |
| Discount $\gamma$ for both rewards | 0.99 | ← | ← | ← | ← |
| Number of updates per training iteration | 10 | ← | 1 | ← | ← |
| Target critic network update rate $\tau$ | N/A | N/A | $5 \times 10^{-3}$ | ← | ← |
| Target critic network update period | N/A | N/A | 1 | ← | ← |
| Replay buffer size | N/A | N/A | $1.6 \times 10^6$ | ← | ← |

Table 6: Hyperparameters used in our experiments of Safety Gym for different approaches. The symbol "←" means the same value with the column on the left. *Both for the policy and value/critic networks. SAC-actor2x-Lag has a double-size policy network. °We normalize each dimension of the reward vector by its moving average mean and standard deviation, and the clipping is performed on normalized values. †This roughly assumes that the target action distribution has a probability mass concentrated on $\frac{1}{10}$ of the support $[-1, 1]$. ‡Following the FOCOPS paper (Zhang et al., 2020). ▷We explored both $10^{-4}$ and $3 \times 10^{-4}$ for PPO/FOCOPS, and the former was selected.

| Hyperparameter | PPO-Lag | FOCOPS | SAC | SAC-actor2x-Lag | SEditor |
|---|---|---|---|---|---|
| Number of parallel environments | 16 | ← | ← | ← | ← |
| Initial rollout steps before training | N/A | N/A | 50000 | ← | ← |
| CNN layers $(channels, kernel\ size, stride)^*$ | $(32, 8, 4), (64, 4, 2), (64, 3, 1)$ | ← | ← | ← | ← |
| Number of hidden layers after CNN$^*$ | 2 | ← | ← | ← | ← |
| Number of hidden units of each layer after CNN$^*$ | 256 | ← | ← | ← | ← |
| Frame stacking | 1 | ← | ← | ← | ← |
| Hidden activation for CNN | relu | ← | ← | ← | ← |
| Entropy regularization weight | $10^{-2}$ | N/A | N/A | N/A | N/A |
| Entropy target per dimension | N/A | N/A | $-1.609^\dagger$ | ← | $(-1.609, -0.916^\dagger)$ |
| Learning rate | $3 \times 10^{-4}$ | ← | ← | ← | ← |
| Mini-batch size | 128 | ← | 256 | ← | ← |

Table 7: Hyperparameters used in our experiments of safe racing for different approaches. The symbol "←" means the same value with the column on the left. $^*$Both for the policy and value/critic networks. SAC-actor2x-Lag has a double-size policy network. $^\dagger$This roughly assumes that the target action distribution has a probability mass concentrated on $\frac{1}{5}$ of the support $[-1, 1]$.

# I Success and Failure Modes

Finally, we show example success and failure modes of SEditor on different tasks in Figure 13 and Figure 14, respectively. We briefly analyze the failure case of each episode in Figure 14. In CARGOAL2, the robot faced a crowded set of obstacles in front of it, making its decision very difficult considering the safety requirement. It took quite some time to drive back and forth, before committing to a path through the two vases in its right front (the fourth frame). However, when passing a vase, the robot incorrectly estimated its shape and the distance to the vase. Even though the majority of its body passed, its left rear tire still hit the vase (the last two frames). In POINTBUTTON2, the robot sped too much in the beginning of the episode, and collided into an oncoming gremlin due to inertia (the floor is slippery!). It did not learn a precise prediction model of the gremlin's dynamics. In CARPUSH2, the robot spent too much time getting the box away from the pillar and did not achieve the goal in time. These failure cases might be just due to insufficient exploration in similar scenarios. In SAFERACINGOBSTACLE, the robot learned to take a shortcut for most sharp turns, essentially sacrificing some utility rewards for being safer (skipping obstacles). The reason is that during every sharp turn with a certain speed, the car's state is quite unstable. It requires very precise control to avoid obstacles during this period, which has not been learned by our approach.

CARGOAL2

CARBUTTON2

CARPUSH2

SAFERACING

SAFERACINGOBSTACLE

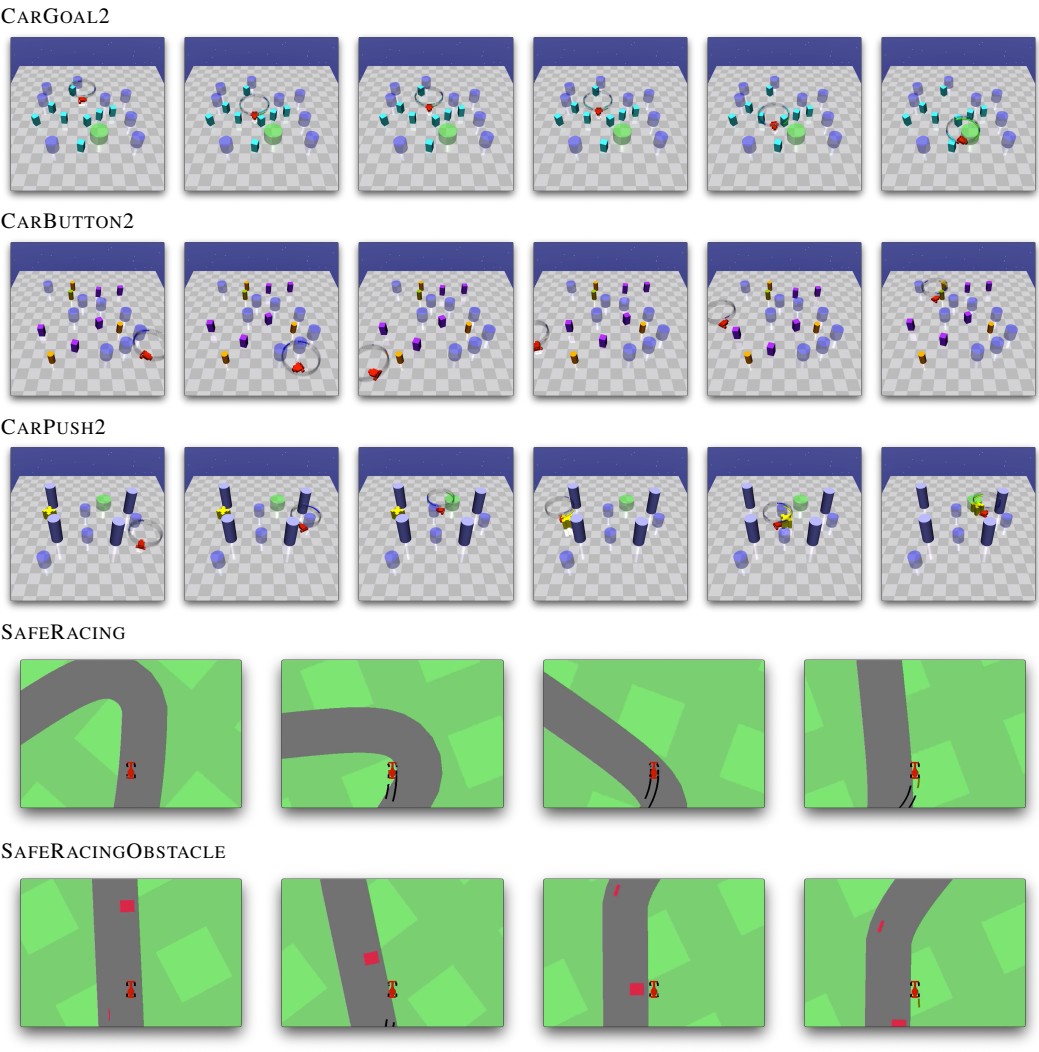

Figure 13: Key frames of several successful episodes of our approach. The robot in each episode finishes the task without violating any constraint. For the safe racing tasks, we only show one representative segment of the track due to space limit.

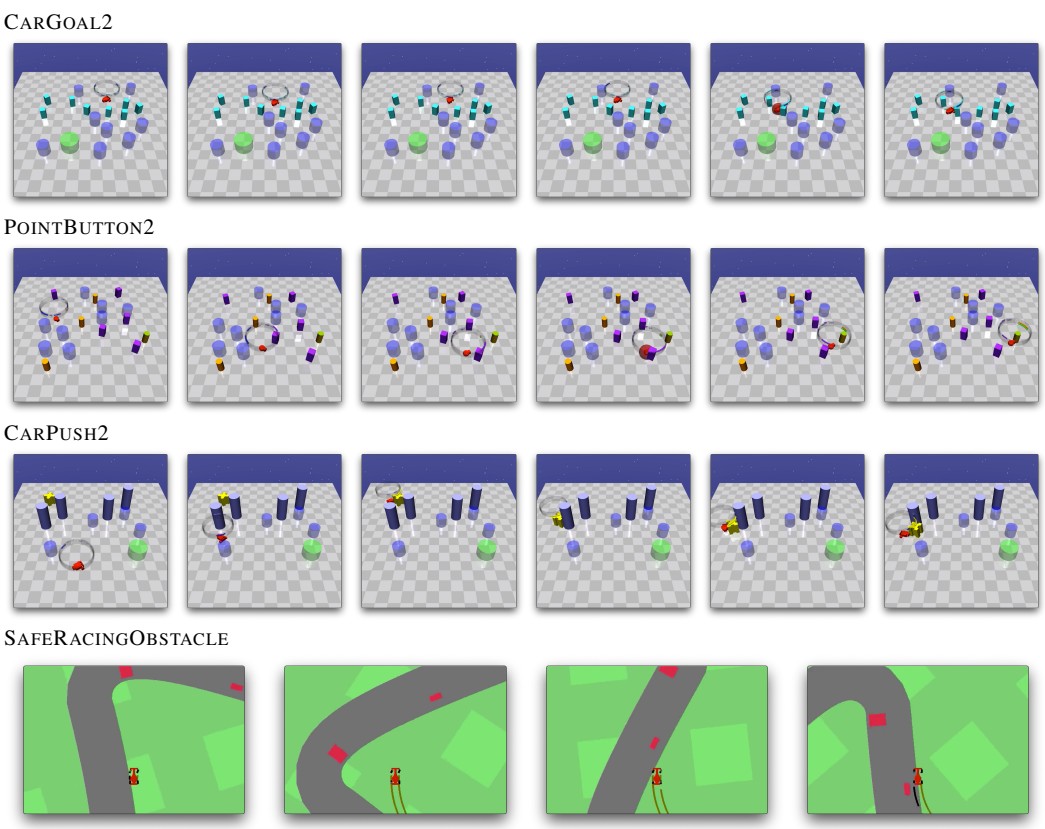

Figure 14: Key frames of several episodes where the robot violated constraints or only learned a sub-optimal policy. In Safety Gym, whenever a constraint is violated, a red sphere is rendered around the robot (the fifth frame of CARGOAL2 and the fourth frame of POINTBUTTON2).

## J   Pseudocode

**Algorithm 1:** SEditor

**Input:** Learning rate $\alpha$
**Initialize:** Randomize $\theta$, $\phi$, and $\psi$; reset the replay buffer $\mathcal{D} \leftarrow \emptyset$
**for** *each training iteration* **do**
> *// Rollout begins*
> Reset the rollout batch $\mathcal{B}_r \leftarrow \emptyset$
> **for** *each rollout step* **do**
> > Action proposal by UM: $\hat{a} \sim \pi_\phi(\hat{a}|s)$
> > Action editing by SE: $\Delta a \sim \pi_\psi(\Delta a|s, \hat{a})$
> > Output action $a = h(\hat{a}, \Delta a)$
> > Environment transition $s' \sim \mathcal{P}(s'|s, a)$
> > Add the transition to the rollout batch $\mathcal{B}_r \leftarrow \mathcal{B}_r \bigcup\{(s, a, s', r(s, a), r_c(s, a))\}$
>
> **end**
> Store the rollout batch in the buffer $\mathcal{D} \leftarrow \mathcal{D} \bigcup \mathcal{B}_r$
> *// Training begins*
> Estimate the gradient of the Lagrangian multiplier $\lambda$ by evaluating Eq. 9 on $\mathcal{B}_r$
> Update the multiplier by Eq. 10: $\lambda_0 \leftarrow \lambda_0 - \alpha\Lambda_{\pi_{\psi\circ\phi}}$
> Sample a training batch $\mathcal{B}$ from the replay buffer $\mathcal{D}$ for gradient steps below
> Perform one gradient step on the critic parameters $\theta$ by TD backup (Eq. 5) on $Q$ and $Q_c$
> Perform one gradient step on UM: $\phi \leftarrow \phi + \alpha\Delta\phi$ (gradient of Eq. 7, a)
> Perform one gradient step on SE: $\psi \leftarrow \psi + \alpha\Delta\psi$ (gradient of Eq. 7, b)
> Update other parameters such as entropy weight, target critic network, etc.

**end**