# OpenReview forum: "Towards Safe Reinforcement Learning with a Safety Editor Policy"
_NeurIPS.cc/2022/Conference — NeurIPS 2022 Accept_

### Official Review · Reviewer_AoPc · 2022-06-20

**Rating:** 7
**Confidence:** 4
**Soundness:** 3 good
**Presentation:** 4 excellent
**Contribution:** 3 good

**Summary:**

The authors consider the problem of safe reinforcement learning which has attracted a lot of attention in the recent years. While there are many algorithms to address the problem, they do typically have some limitations. One of the major concerns is that the algorithms are quite brittle and require significant tuning  as well as many interactions with the environment to solve the problem. The authors propose a modification of a recently proposed safety layer to solve some of the issues with safe RL.

The learning process is decomposed into three independent learning processes: a) scheduling the Lagrangian multiplier weighing  the estimated cost b) learning the utility policy maximizing the reward, and c) learning the projection onto the safety state. While the learning process a) is identical to the classical procedure for learning the  Lagrangian multiplier, the process b) and c) are novel.  Albeit, there were similar approaches in the past (e.g., [Mguni et al 2021])

The authors then conduct a number of experiments showcasing the performance of their approach in the case of small cost limits.

**Questions:**

1. While the motivation for the hinge loss makes sense, estimates of the value functions can be quite noisy. I wonder if the authors made some steps to alleviate this issue. Further, would it make sense to consider the $l_2$ distance between the $Q$ functions instead of the hinge loss? Does it work worse?

2. I suggest expanding the motivation behind learning the policy $\pi_\psi(\cdot | s, a)$ predicting the update $\delta a$ instead of learning the policy predicting the corrected action $\hat a$, i.e.,  $\hat a \sim \pi_\psi(\cdot | s, a)$.  That is, I recommend moving parts of the Appendix D into the main text.

3. The choice of nonlinearity $h$ is a bit confusing, why $\delta a$ is multiplied by two?

4. Is it possible to elaborate if the correction policy takes into account the long term behavior of safe actions. This could be important since the safety layer by Dalal et al makes the correction decisions based on the instantaneous information (costs, states and actions)

5. Please add absolute values for the returns in order to compare the performance to other works. Normalizing the returns unfortunately makes such a comparison impossible.

6. While having an almost zero cost limit is an important problem to consider, it is important to compare the performance on the benchmarks as they were intended. Hence adding some experiments (say point push and/or car push) with the cost limit of $25$ is very important.

7. It seems that the proposed improves sample efficiency of the Lagrangian SAC, but doesn't really improve the performance in many tasks. Hence, I suggest running the experiments for longer steps for SAC to make sure that the performance is indeed worse.

8. The baselines are not unfortunately the latest in safe RL. I suggest comparing to [Stooke et al 2020]  and [As et al 2022]. The latter reference published numerical values of their experiments with the code. One could also consider [Liu et al 2022], but it was just recently published.

9. Could the approach work with a pre-trained utility policy? That is, making an already existing policy safe.

Missing references:
* [Stooke et al 2020] Stooke, Adam, Joshua Achiam, and Pieter Abbeel. "Responsive safety in reinforcement learning by PID Lagrangian methods." International Conference on Machine Learning. PMLR, 2020.
* [As et al 2022] As, Yarden, et al. "Constrained Policy Optimization via Bayesian World Models." ICLR 2022
* [Mguni et al 2021] Mguni, David, et al. "DESTA: A Framework for Safe Reinforcement Learning with Markov Games of Intervention." arXiv preprint arXiv:2110.14468 (2021)
* [Liu et al 2022] Liu, Zuxin, et al. "Constrained variational policy optimization for safe reinforcement learning." arXiv preprint arXiv:2201.11927 (2022)

NB: I would consider raising my score if some of these questions are answered, especially the concerns relating to the experiments section.

**Limitations:**


1. Providing some intuition would help to understand the approach better.
2. I am not sure if anything can be done for the theoretical analysis, but it would be a quite welcome addition as well.
3. It is necessary to consider the case of large cost limits.
4. As discussed above, comparison to more baselines and the presentation of the existing results should be improved

**Strengths And Weaknesses:**

The paper is well-written and the presentation is great and has only a few minor problems. The experiments look impressive as well. Overall, I enjoyed reading this paper, but there are a few weaknesses in my point of view:

1. Some of the design choices are not very clear. For example. it is not clear why separating the policy learning process into two makes sense. There's naturally some empirical evidence, but the splitting process is not very intuitive. I suggest providing an intuition why this approach works and how.

2. Theoretical analysis is limited. In particular, I am a bit concerned that there's a an implicit assumption that transition between a policy maximizing the reward and a safe policy is continuous.

3. Presentation of the experiments is quite impressive, but a few improvements are necessary (see questions).

---

> ### Author Response · Authors · 2022-08-02
> **Replies to Reviewer AoPc (part 1/3)**
>
> > Some of the design choices are not very clear. For example. it is not clear why separating the policy learning process into two makes sense. There's naturally some empirical evidence, but the splitting process is not very intuitive. I suggest providing an intuition why this approach works and how.
>
> Please see our rebuttal summary (1) for a detailed response to this question.
>
> > I am a bit concerned that there's a an implicit assumption that transition between a policy maximizing the reward and a safe policy is continuous.
>
> SE outputs safe actions by editing in a bounded action space, and according to our editing function $\min(\max(\hat{a} + 2\Delta a, -1),1)$, SE in theory has _total_ control of the output action (whose range is $[-1,1]$) regardless of the action proposal $\hat{a}$. It can completely change $\hat{a}$ in whatever way that is safe. For example, if $\hat{a}$ says 'turn left' (-1), but SE realizes that only 'turn right' (+1) is safe, then it can output $\Delta a=1$ to overwrite $\hat{a}$. Thus SEditor doesn't have an implicit assumption of continuous transition between a policy maximizing the reward and a safe policy.
>
> > While the motivation for the hinge loss makes sense, estimates of the value functions can be quite noisy. I wonder if the authors made some steps to alleviate this issue.
>
> We didn't have special techniques to address the noise in the values. Even though in the beginning the values are noisy, we have an entropy regularization term that encourages both UM and SE to explore randomly. The weight for this entropy term slowly decreases over time. This somewhat decreases the impact of value noises in the beginning. As the training goes, the values become reasonably accurate, as illustrated in Figure 12.
>
> > Further, would it make sense to consider the distance between the functions instead of the hinge loss? Does it work worse?
>
> The distance between functions forces the utility Q values to be the same for $\hat{a}$ (before editing) and $a$ (after editing). However, we actually don't want to penalize the case $Q(s,a)>Q(s,\hat{a})$. For a simple example, suppose $Q(s,\hat{a})=1$ and $\lambda=1$, and consider two edited action values:
>
> |$a$ |$Q(s,a)$ |$Q_c(s,a)$|$-(Q(s,a)-Q(s,\hat{a}))^2+\lambda Q_c(s,a)$|$-\max(0,Q(s,\hat{a})-Q(s,a))+\lambda Q_c(s,a)$|
> |:-------:|:----------:|:--------:|:---------:|:-----------:|
> |$a_1$|$1$|$-0.2$ |$-0.2$ (better) |$-0.2$ (worse)|
> |$a_2$|$2$|$-0.1$ |$-1.1$ (worse)  |$-0.1$ (better)|
>
> We see that even though $a_2$ are better regarding both utility and safety critics, with L2 distance it actually results in a lower objective value.
>
> > That is, I recommend moving parts of the Appendix D into the main text.
>
> We will expand more on why we choose to predict $\Delta a$ for SE in the main text, probably with one subfigure of Figure 12 moved there to help the explanation.
>
> > The choice of nonlinearity $h$ is a bit confusing, why $\delta a$ is multiplied by two?
>
> Since the action proposal $\hat{a}\in[-1,1]$ and action change $\Delta a\in[-1,1]$, in order to let SE have full control of outputting $a$ in $[-1,1]$, we multiply $\Delta a$ by two. Otherwise, for some values such as $\hat{a}=-1$, $\min(1,\max(-1,\hat{a}+\Delta a)) \in [-1, 0]$ which cannot cover the whole space of $[-1,1]$.
>
> > Is it possible to elaborate if the correction policy takes into account the long term behavior of safe actions. This could be important since the safety layer by Dalal et al makes the correction decisions based on the instantaneous information (costs, states and actions)
>
> According to the SE loss in Eq 7b, the safety policy does take into account the long-term safe actions, because it tries to maximize the safety critic which is learned to be the expected future return of constraint rewards. This is a major difference between our safety editor and the safety layer in [Dalal et al 2018].

---

> ### Author Response · Authors · 2022-08-02
> **Replies to Reviewer AoPc (part 2/3)**
>
> > Please add absolute values for the returns in order to compare the performance to other works. Normalizing the returns unfortunately makes such a comparison impossible.
>
> Note that our safety gym environment has been customized to equip the agent with a more advanced lidar sensor in the following two ways:
>
> 1) It is a natural lidar instead of a pseudo lidar adopted by the original benchmark in [Ray et al 2019]. The natural lidar simulates a ray intersecting with an object, starting from an origin. The pseudo lidar can only detect object centers. Natural lidar reveals shape information of objects, which is required to achieve a very low constraint threshold.
>
> 2) Our lidar has 64 bins while the original lidar in [Ray et al 2019] has only 16 bins. The 16-bin pseudo lidar's precision is too low and is actually a bottleneck of achieving much fewer safety violations. Without perception precision, the problem is ill-posed for a robot suffering from blind spots to avoid touching objects.
>
> Moreover, to make the utility metric more interpretable by humans and also to make its scale less affected by the map size and task type, we report success rates instead of raw returns (success means the robot achieves the goal before timeout).
>
> Thus our result is not directly comparable to existing works that use the original safety gym environment.
>
> > While having an almost zero cost limit is an important problem to consider, it is important to compare the performance on the benchmarks as they were intended. Hence adding some experiments (say point push and/or car push) with the cost limit of 25 is very important.
>
> Our environment has been modified as explained in the previous reply. Therefore, the benchmarks are no longer the same with [Ray et al 2019] even with the original cost limit of 25. We've run SEditor and SAC-actor2x-Lag with a cost limit of 25 on PointPush-{1,2} and CarPush-{1,2} for 5M steps. The training curves are at https://i.imgur.com/leSm6Zt.png. The SWU scores are:
>
> | |CP1|CP2|PP1|PP2|
> |-----:|-----|-----|-----|-----|
> |SAC-actor2x-Lag|0.98|1.27|1.46|0.39|
> |SEditor|1.01|1.90|1.61|3.74|
>
> We see that SAC Lagrangian still struggles in the Push tasks even with a much higher cost limit.
>
> > It seems that the proposed improves sample efficiency of the Lagrangian SAC, but doesn't really improve the performance in many tasks.
>
> First of all, on safe racing tasks, it is clear that SAC-actor2x-Lag has worse performance (not just sample efficiency) than SEditor.
>
> Second, the safety gym curves in Figure 3 can only reveal large gaps in the utility performance. It is somewhat difficult to notice the difference between safety performance as the violation rates are in a range of small values. For example, if one method gets a violation rate of 1e-3 and another one gets a rate of 5e-4, their visual difference is very tiny on the curves even though one method has 100% more violations than the other.
>
> That's the reason why we also report SWU scores in Table 1. From it (the updated one below with 9 seeds), SAC-actor2x-Lag is clearly worse than SEditor on CP1, CG1, CP2, PP1, PP2, and PB2, among which CG1 and PB2 have obvious gaps in the scores but tiny *visual* gaps in the safety curves.
>
> |    |CP1|CG1|CB1|CP2|CG2|CB2|PP1|PG1|PB1|PP2|PG2|PB2|Overall|Improvement|
> |------:|-----|-----|-----|-----|-----|-----|-----|-----|-----|-----|-----|-----|----|----|
> |SAC-actor2x-Lag|0.74|0.63|0.70|0.59|1.00|0.81|0.60|1.00|0.89|0.37|0.94|0.64|0.74|51\%|
> |SEditor|**1.01**|**0.94**|0.78|**1.49**|0.99|**0.95**|**1.55**|1.00|**1.00**|**1.78**|1.00|**1.00**|**1.12**|-|
>
> According to the definition, two SWU scores are best compared by division. So SEditor is doing about $1.12/0.74-1=51$\% better than SAC-actor2x-Lag on Safety Gym.

---

> ### Author Response · Authors · 2022-08-02
> **Replies to Reviewer AoPc (part 3/3)**
>
> > Hence, I suggest running the experiments for longer steps for SAC to make sure that the performance is indeed worse.
>
> As requested, we report here the SWU score comparison for SAC-actor2x-Lag and SEditor with 10M steps on Safety Gym. (The scores are calculated based on the performance of unconstrained SAC at 10M steps, and are not comparable to scores at 5M steps.)
>
> |    |CP1|CG1|CB1|CP2|CG2|CB2|PP1|PG1|PB1|PP2|PG2|PB2|Overall|Improvement|
> |------:|-----|-----|-----|-----|-----|-----|-----|-----|-----|-----|-----|-----|----|----|
> |SAC-actor2x-Lag|0.91|0.84|**1.00**|0.82|0.84|0.80|1.02|1.00|0.74|0.75|1.00|1.00|0.89|17\%|
> |SEditor|1.01|**1.00**|0.85|**1.28**|**1.00**|**0.98**|0.94|1.00|**0.97**|**1.42**|1.00|1.00|**1.04**|-|
>
> Training more steps somewhat decreases but not closes the gap between SAC-actor2x-Lag and SEditor regarding final performance.
>
>
> > The baselines are not unfortunately the latest in safe RL. I suggest comparing to [Stooke et al 2020] and [As et al 2022]. The latter reference published numerical values of their experiments with the code. One could also consider [Liu et al 2022], but it was just recently published.
>
> The main contribution of this paper is a two-policy framework that decomposes the maximization of utility and safety into two easier subtasks. As a first-order approach, we directly adopt the classic Lagrangian method to adjust the weight between utility and safety. [Stooke et al. (2020)] is kind of orthogonal to our focus here, in that it improves the dynamics of the Lagrangian multiplier with PID control (Algorithm 2 in their paper), addressing a separate issue of oscillation as mentioned in our limitations section. It is complementary to our approach and can be integrated into our approach. We believe that it will decrease the oscillation locally but still have a similar global trend of $\lambda$ like ours.
>
> [As et al 2022] considers a subset of 6 safety gym tasks (mostly level-1). It assumes RGB image inputs in order to use Bayesian world models. Thus their published numerical results are not directly comparable to ours, nor can the code be directly applied to our lidar inputs.
>
> [Liu et al 2022] also considers safety gym with a 16-bin pseudo lidar, on a subset of 5 tasks (mostly level-1). Moreover, they modified the environment to have a fixed layout across episodes instead of randomized ones, to decrease the training difficulty. In contrast, we use 12 tasks (both level-1 and level-2) with randomized layouts. Given the limited time frame of rebuttal, we were unable to adapt their code to our tasks with low constraint thresholds.
>
> We will add [As et al 2022] and [Liu et al 2022] to our Related Work section.
>
> > Could the approach work with a pre-trained utility policy? That is, making an already existing policy safe.
>
> A pre-trained utility policy is accepted by our approach only if it is allowed to be fine-tuned. In Eq 7a, UM maximizes its objective *through the lens* of SE. It knows how to adjust its output $\hat{a}$ given its expectation of SE's behavior. The gradient of $\phi$ will flow through SE $\pi_{\psi}$. In other words, UM's MDP has been changed by SE. For a pre-trained utility policy, it has to be fine-tuned to adapt to the behaviors of SE.
>
> > Missing reference:
>
> Thanks for pointing out the missing references and we will cite them in the revision. ([Stooke et al 2020] has already been cited in the paper.)

---

> ### Comment · Reviewer_AoPc · 2022-08-04
> **further clarifications and suggestions**
>
> I thank the authors for the replies and clarifications. I am happy with the answers, their design choices are well explained, and the theoretical questions have been answered.
>
> I would suggest discussing possible use of pre-trained policies and long term safe action correction in more detail in the paper (but maybe I missed some details there).
>
> Before I determine the final score improvement, I would like (again) to request comparing to [Stooke et al 2020] on the unmodified PointGoal1. Note that [Stooke et al 2020] experimented on PointGoal1, if I am not mistaken, hence tuning the baseline shouldn't be a problem.

---

> > ### Author Response · Authors · 2022-08-05
> > **Additional experiment on the unmodified PointGoal1**
> >
> > > I would suggest discussing possible use of pre-trained policies and long term safe action correction in more detail in the paper (but maybe I missed some details there).
> >
> > We've added related discussions to Section 3 of the revised manuscript.
> >
> > > Before I determine the final score improvement, I would like (again) to request comparing to [Stooke et al 2020] on the unmodified PointGoal1. Note that [Stooke et al 2020] experimented on PointGoal1, if I am not mistaken, hence tuning the baseline shouldn't be a problem.
> >
> > We've run SEditor (learning rate for $\lambda$ is 1e-2) on the unmodified PointGoal1. The training curves (aggregated over 4 seeds) are https://i.imgur.com/ZQeW9wk.png. It's unsurprising that SEditor did pretty well under a high cost limit.
> > Some explanations about the result:
> > 1. Our violation rate threshold 0.025 is equivalent to the original cost limit 25 (per episode) divided by the episode length 1000.
> > 2. In Appendix C.1 Figure 1 of [Stooke et al 2020], the utility performance under various learning rates ($K_I$) is about the same, with slight differences in sample efficiency. Some comparison between their result and ours at different environment steps:
> >
> > |Steps | [Stooke et al 2020]| SEditor|
> > |-------|-----------------------|----------|
> > |2.5e7 | 26 | 29|
> > |1.2e7 | 24 | 27|
> > |6e6 | 22 | 25|
> >
> > 3. Without P-control, our cost curves are similar to the ones with $K_P=0$ in [Stooke et al 2020], which is expected: the initial cost was high and then quickly dropped to the limit. Our cost stabilized at about 3M steps while [Stooke et al 2020] stabilized at about 10M steps for $K_I=1e-2$.
> >
> > Again, we emphasize that this is not an apples-to-apples comparison between SEditor and [Stooke et al 2020], since there are lots of different variables in the two setups.

---

> > > ### Comment · Reviewer_AoPc · 2022-08-07
> > > **final remarks**
> > >
> > > I agree that there are always different aspects of the algorithms and presenting a comparison on the fly (within a week) is not fair.
> > > I actually suspected SEditor would be less effective in this setting, but these learning curves exceeded my expectations. Thanks a lot for adding these in the rebuttal!
> > >
> > > I raised my scores.

---

### Official Review · Reviewer_iJnb · 2022-07-05

**Rating:** 6
**Confidence:** 4
**Soundness:** 3 good
**Presentation:** 3 good
**Contribution:** 2 fair

**Summary:**

This paper studies the problem of safe reinforcement learning in which the learning agent seeks to optimize the reward while minimally reducing the number of cost violations. The paper points out that there are two main commonly used assumptions in the field: the oracle function for verifying whether the state is safe or not, and the pre-trained safe policy that brings the agent to a safe state. However, these approaches make the algorithm hard to generalize to other applications. In this paper, the assumption is that the agent only sees whether the state is safe or not “only after” it visits that state. This makes the problem challenging and the learning agent needs to learn to be safe from scratch. As a result, the paper says that we can only satisfy the constraint “asymptotically”. The paper proposes a two-policy structure safe reinforcement learning algorithm: a utility maximizer (UM) that only optimizes the reward function without considering the safety constraints, and a safe editor (SE) that transforms the action proposed by UM into a safe action. SE is also a reinforcement learning agent with the goal to maximize the “constraint” reward while forcing the state-action values to be similar before and after editing. There are several contributions: (1) the proposed algorithm does not require an inner loop to edit the action to the safer one, which makes the algorithm easy to use; (2) they show that hinge loss is better than L2 when constraining the distance of actions before and after editing, and (3) they verify the algorithm in the experiment. The experiment results are as follows: (1) Table 1 and Figure 3 show the results of SWU scores and the learning curves over the number of steps. The paper points out that SEditor has a higher score than the baselines on 7 out of 12 tasks. (2) Figure 4 shows the visualization of rollout trajectories at different training stages. They show that SEditor is able to find the feasible path without being blocked, and finally (3) Figure 5 shows the result in the racing task.

**Questions:**

(1) In Figure 4, could you give the reader more intuition on why SEditor has a better exploration strategy?
(2) Other than empirical observations, could you explain why hinge loss is better than L2 loss theoretically?
(3) After we train UM, do we need SE during deployment. The reader thinks the answer is yes since UM never learns how to be safe. If this is the case, what is the absolute value of the $\delta a$ along the training? I expect the $|\delta a|$ would be the same across training. The reader is curious to see this plot.
(4) How to apply the proposed algorithm in real-world application? For example, learning a policy for self-driving cars.
(5) Please address the comment on number 2.

I am happy to increase the score if the author answers the question well. Thank you.


**Limitations:**

See the cons in the previous section.

**Strengths And Weaknesses:**

Pros:
(1) The paper is well-written.
(2) The assumption presented in the paper is clear. For example, it clearly says that we can only learn to be safe asymptotically due to the lack of the oracle that tells whether the state is safe or not. This point is often missed in the literature.
(3) The literature review is good. It covers the work from safe reinforcement learning, especially the work with a two-policy structure.

Cons:
(1) As pointed out by the paper, the proposed approach has some similarities to recovery RL. The difference seems to be that recovery RL switches hardly from the learner policy to the safe policy whereas, in this paper, SE edits the action from UM by adding.
(2) I generally agree that the method here is simple and effective based on the experiment. But there is no discussion about the failure case of the proposed algorithm. One possible failure case of the proposed algorithm is that the action proposed by UM may exceed the editing capability of SE. For example, if the learning agent enters the state called “near-termination state”, meaning that the agent has not died, but every possible action is no longer safe enough to recover the agent from this state. Then the proposed approach could fail.
(3) There is no theory in the paper to support the claim that the agent will learn safely.

---

> ### Author Response · Authors · 2022-08-02
> **Replies to Reviewer iJnb (part 1/2)**
>
> > As pointed out by the paper, the proposed approach has some similarities to recovery RL. The difference seems to be that recovery RL switches hardly from the learner policy to the safe policy whereas, in this paper, SE edits the action from UM by adding.
>
> Aside from this technical difference, another major difference in the method assumptions is that recovery RL assumes an offline dataset of demonstrated safety trajectories (which could be difficult to obtain) to be provided for learning the recovery policy first before the downstream tasks are trained. The recovery policy is _fixed_ throughout the second-stage training. As a result, it is necessary that the offline dataset has the same safety constraints with the downstream tasks in order for the recovery policy to work well. We don't require a pretraining stage from demonstrations and our SE policy can evolve together with UM.
>
> > I generally agree that the method here is simple and effective based on the experiment. But there is no discussion about the failure case of the proposed algorithm.
>
> We actually discussed several failure cases in Appendix G. In summary, it's still difficult for the agent to learn a precise safety critic or (implicitly) a precise prediction of the dynamics of other moving objects. This can be due to insufficient exploration or the inherent lack-of-planning nature of SEditor being model-free. Note that these failure cases are not surprising considering our very harsh safety thresholds in the experiments.
>
> > One possible failure case of the proposed algorithm is that the action proposed by UM may exceed the editing capability of SE. For example, if the learning agent enters the state called “near-termination state”, meaning that the agent has not died, but every possible action is no longer safe enough to recover the agent from this state. Then the proposed approach could fail.
>
> Actually, this "failure case", from a theoretical perspective, is unlikely to appear in our approach. This scenario is about whether our safety editor is shortsighted so that it doesn't stop the agent entering non-recoverable states until it's too late. For a safety layer such as the one in [Dalal et al 2018] that only looks at immediate safety costs, this could happen. However, our loss to train the safety editor (SE) in Eq 7b uses a *safety critic that consider future constraint rewards*. If there are indeed non-recoverable states in which the SE can't do anything about the safety, the safety critic will train SE to modify the actions long before that happens.
>
> In other words, our approach doesn't modify the action proposal by looking at immediate safety costs, instead it considers the long-term cost. Theoretically, it will stop the agent entering non-recoverable states in the first place.
>
> > There is no theory in the paper to support the claim that the agent will learn safely.
>
> Because we don't assume access to an oracle of safety model, our agent can only learn to be safe asymptotically. All safety knowledge has to be obtained online by interacting with the environment. Therefore, we didn't claim in the paper that "the agent will learn safely".
>
> > In Figure 4, could you give the reader more intuition on why SEditor has a better exploration strategy?
>
> According to our rebuttal summary, SEditor has a feature of "guarded exploration".
>
> From the perspective of UM, its MDP (precisely, action space) is altered by SE. UM's actions are guarded by the barriers set up by the SE. Instead of discouraging an unsafe action by UM (i.e., punishing the robot with negative signal), SE actually gives suggestions to UM by redirecting the unsafe action to a safe but also utility-high action to continue UM's exploration. This guarded exploration leads to a better overall exploration strategy because safety constraints are less likely to hinder UM's exploration (Figure 4 lower right).
>
> In contrast, SAC-Lag couples the learning of utility and safety together. Any unsafe action will be punished with a low safety critic value. This makes the policy conservative and greatly hinders its exploration: the trajectories are confined in a small region (Figure 4 lower left).
>
> We will add more explanations to Figure 4.

---

> ### Author Response · Authors · 2022-08-02
> **Replies to Reviewer iJnb (part 2/2)**
>
> > Other than empirical observations, could you explain why hinge loss is better than L2 loss theoretically?
>
> L2 loss minimizes the Euclidean distance between $\hat{a}$ (before editing) and $a$ (after editing). This L2 distance is a measurement of how close $\hat{a}$ is to $a$ in the action space, and is only a very rough estimate of their closeness in their Q value space (approximation to the Taylor series of Q). What we really care about is how the Q value changes after the action is edited.
>
> As an example, let's suppose that the agent faces obstacles in the front. UM outputs $\hat{a}=-1 (left)$ but SE modifies it to $a=1 (right)$ because of the potential safety issue on the left side. With the hinge Q loss, SE won't be penalized as long as the utility Q value is not decreased after editing. In fact, 'right' makes the agent achieve the goal no slower than 'left' does. So 'left' and 'right' are functionally similar regarding goal achievement even though they are far away in the action space. The L2 action loss ignores the utility of actions and could affect SE's decision in a bad way.
>
> > After we train UM, do we need SE during deployment. The reader thinks the answer is yes since UM never learns how to be safe. If this is the case, what is the absolute value of the $\delta a$ along the training? I expect the $|\delta a|$ would be the same across training. The reader is curious to see this plot.
>
> Yes, during deployment we still need SE.
>
> Note that $|\Delta a|$ alone doesn't reflect the actual editing because the editing function is nonlinear w.r.t. $|\Delta a|$. Recall that the editing function is $a=\min(\max(\hat{a}+2\Delta a),-1),1)$. For $\hat{a}=1$, even if SE outputs $\Delta a=1$, the edited action $a$ is still 1 due to clipping. So it's only meaningful to look at $|a - \hat{a}|$.
>
> This value is not the same across training. A simple fact is that in the beginning of training, SE hasn't learned how to edit any unsafe action yet and its output $\Delta a$ could be random for exploration. After some training time, SE learns the fact that most action proposals $\hat{a}$ by UM is already safe and $|a-\hat{a}|$ becomes smaller. As a representative example, the plot of $|a-\hat{a}|$ for SafeCarPush1 across training is at (action dim = 2): https://i.imgur.com/Afnmfg2.png.
>
> > How to apply the proposed algorithm in real-world application? For example, learning a policy for self-driving cars.
>
> Since SEditor can only learn to behave safely in an asymptotic manner, it will violate safety constraints during training. This might be problematic for real-world applications because the real-world consequences are usually more costly than those in simulators. However, there are two strategies with which SEditor can be applied:
>
> 1.  sim-to-real transfer. If the simulator is complex enough and with domain randomization, sim-to-real transfer is a possible way of using SEditor. After achieving an extremely low violation rate in the simulator, we deploy SEditor to real world.
>
> 2.  Extra protection of the robot and scene during training. For example, better-quality materials are used for training robots in case of damages, compared to the deployed robots.
>
> > Please address the comment on number 2.
>
> We assume that here "number 2" refers to "Cons (2)". Again we would like to address that unlike [Dalal et al. 2018], our SE edits actions depending on a long-term constraint return instead of immediate constraint rewards. Thus it can prevent the agent entering non-recoverable states.

---

> > ### Comment · Reviewer_iJnb · 2022-08-08
> > **Thank you**
> >
> > Dear Authors,
> > Thank you for the rebuttal. After reading the rebuttal and the reviews, I have updated the score. Thank you.

---

### Official Review · Reviewer_hrVA · 2022-07-07

**Rating:** 7
**Confidence:** 4
**Soundness:** 3 good
**Presentation:** 3 good
**Contribution:** 4 excellent

**Summary:**

This paper proposes a new and general approach for safe RL with relatively strong empirical evidence. The proposed approach leverages two policies, the first one which is a utility maximizer (UM), and the second one which is a safety editor (SE) in charge of adjusting the UM action to ensure a sufficiently low constraint violation rate. The method is compared against existing safe RL algorithms in several continuous control tasks. The experiment section also shows some ablation studies (in Appendix).

**Questions:**

I find a few issues or have some comments with the paper, please see below:

1. Have the authors considered using a parametric action editing (h) function? To be clear, I think this is a reasonable choice, but could the authors elaborate on what motivated the non-paremetric choice?
2. The proposed approach seem to introduce structure into the policy search by decomposing the safe RL task into two subtasks, the reward maximisation subtask (governed by \phi) and the safety subtask (governed by /psi). Could the authors elaborate on the differences with the mechanism introduced in [2] where the safety subtask is given to an adversary and the actor is given the subtask to maximize future expected rewards while also maximizing its discrepancy with the adversary. To what extent is the policy search space different? I note that the paper [2] is quite recent (April 20th 2022). That being said, a short discussion highlighting the similarities and differences may help to give further context.
3. Additional relevant work: [3].
4. I understand that the number of tasks is high but I would kindly encourage the authors to use resource constraints on presenting more seeds with less tasks than the contrary. Unfortunately, using 3 seeds in deep RL cannot be statistically conclusive.

Mainly based the last point, my score is what it is, but I will significantly increase it if the above concerns/comments (in particular the last one) are addressed.

[1] Henderson, P., Islam, R., Bachman, P., Pineau, J., Precup, D., & Meger, D. (2018, April). Deep reinforcement learning that matters. In Proceedings of the AAAI conference on artificial intelligence (Vol. 32, No. 1).
[2] Flet-Berliac, Y., & Basu, D. (2022). SAAC: Safe Reinforcement Learning as an Adversarial Game of Actor-Critics. In 5th Conference on Reinforcement Learning and Decision Making.
[3] Bhatnagar, S., & Lakshmanan, K. (2012). An online actor–critic algorithm with function approximation for constrained markov decision processes. Journal of Optimization Theory and Applications, 153(3), 688-708.

**Limitations:**

Some limitations of the work have been addressed in the conclusion with potential interesting future work directions.

**Strengths And Weaknesses:**

The contribution of this paper is original and appears relevant to the community. Overall the paper is mostly well written and develops its idea clearly. The paper presents an idea that is simple and leads to favorable empirical results. However, the very low number of seeds considered in the “Safety Gym tasks” is very low and does not allow to statistically draw any empirical conclusion [1] for this part. We develop more points below.

---

> ### Author Response · Authors · 2022-08-02
> **Replies to Reviewer hrVA**
>
> > Have the authors considered using a parametric action editing (h) function? To be clear, I think this is a reasonable choice, but could the authors elaborate on what motivated the non-paremetric choice?
>
> If the editing function $h$ is parametric, most likely its parameters have to be learned by the same SE loss (Eq 7b), too.
>
> If this is the case, we can always move the parametrization from the editing function to SE but adopting a simple non-parametric editing function. Given that our SE can be arbitrarily complex, its output $\Delta a$ could depend on the current state $s$ and the action proposal $\hat{a}$ in an arbitrarily complex way. Thus even though the additive operation is simple and non-parametric, the overall editing process is already general enough to represent any modification.
>
> So far we don't have another good loss to train a parametric editing function if Eq 7b is not used for it, and we don't have a clear motivation of moving some of the parameterization of SE to the editing function.
>
> > Could the authors elaborate on the differences with the mechanism introduced in [2] where the safety subtask is given to an adversary and the actor is given the subtask to maximize future expected rewards while also maximizing its discrepancy with the adversary. To what extent is the policy search space different? I note that the paper [2] is quite recent (April 20th 2022). That being said, a short discussion highlighting the similarities and differences may help to give further context.
>
> [Flet-Berliac and Basu 2022] also has a two-policy design. However, their two policies are adversarial while ours are cooperative. More specifically, one of their policies tries to intentionally *maximize* risk (i.e., minimizing constraint rewards in our terminology) by behaving *unsafely*, and "thus to shrink the feasibility region of the agent’s value function". It's essentially asking the question: "What should the agent do if at any state a very risky policy could take over?" Their overall policy will bias towards being conservative. Our SE minimizes risk (maximizing constraint rewards), and UM and SE always *cooperate* to resolve the conflict between utility and safety.
>
> > Additional relevant work: [3].
>
> Thanks for pointing out this work. We'll add to the revision.
>
> > I understand that the number of tasks is high but I would kindly encourage the authors to use resource constraints on presenting more seeds with less tasks than the contrary. Unfortunately, using 3 seeds in deep RL cannot be statistically conclusive. Mainly based the last point, my score is what it is, but I will significantly increase it if the above concerns/comments (in particular the last one) are addressed.
>
> Please see our updated curves and scores with 9 random seeds in Rebuttal Summary (3). Please kindly consider increasing the score if you think the last concern has been well addressed. Thank you!

---

> > ### Comment · Reviewer_hrVA · 2022-08-04
> > **Thank you to the authors and kind request for uploading the revised version**
> >
> > I thank the authors for their response and the time spent adding additional seeds to the experiments.
> >
> > Could the author upload the revised version of the paper when it is ready?
> >
> > After that, I will be happy to update my score to reflect the response.

---

> > > ### Author Response · Authors · 2022-08-05
> > > **Uploaded the revised version**
> > >
> > > We've uploaded the revision according to our responses. Please take a look. Thanks.

---

> > > > ### Comment · Reviewer_hrVA · 2022-08-07
> > > > **Updated my score**
> > > >
> > > > Thanks, had a look at the revised version. Just updated my score.

---

### Official Review · Reviewer_MUd3 · 2022-07-17

**Rating:** 7
**Confidence:** 4
**Soundness:** 3 good
**Presentation:** 3 good
**Contribution:** 3 good

**Summary:**

This paper proposes a new method for tackling reinforcement learning with safety constraints with a two-policy approach. The first policy learns to maximize the overall return while the second policy edits actions coming from the first policy to ensure safety. The new algorithm was evaluated on the SafetyGym tasks and two additional safe racing tasks.

**Questions:**

One additional question I have aside from the points I mentioned in the main review. In discussing the limitations of your work, you mentioned that oscillation is still an issue that you observe, I wonder if you have considered or whether it's possible to integrate some of the ideas from Stooke et al. (2020) (referenced in your paper) which addresses the oscillation issue but for Lagrangian methods.

**Limitations:**

Yes, see above comment.

**Strengths And Weaknesses:**

I like this paper a lot, the idea presented by the authors is simple and elegant yet appears to be quite powerful in practice. The paper is very clearly written and a pleasure to read. The authors clearly explained their methodology and motivated the different design choices they employed (e.g. Hinge loss vs. L2, additive edit function etc.). Experiments as well as detailed ablation studies are very well done and strongly support the authors' arguments. I think this paper makes a strong contribution to the field of safe RL and recommend this paper for acceptance at this venue. Additional detailed comments are as follows:
- The paper argues why the additive edit function is preferred and the authors provide some solid observations for why this is the case. I was wondering if you could give a bit more intuition on why this is the case (especially why additive is better than overwrite).
- Also a related question to the above, have you considered other forms of edit functions aside from additive and overwrite?
- My biggest concern is with how $\Lambda$ is evaluated. My understanding is that $\Lambda$ is evaluated in an on-policy Monte Carlo fashion even though the overall algorithm is off-policy. I agree with the author's argument since off-policy evaluation is quite challenging and generally inaccurate, however I do have several questions:
  - When you use a batch of size $N$, you mentioned that you don't necessarily wait for the episode to finish when doing evaluation. Does this mean each batch could contain multiple episodes or a partial unfinished episode?
  - In general, larger $N$ is preferred from an accuracy perspective but is problematic in terms of sample efficiency. From a theoretical perspective you could get away with a smaller $N$ if the underlying MDP has a small mixing time. However I wonder in practice how did you chose $N$?
  - Also for evaluating $\Lambda$, did you have to collect a separate batch of data or was it the same data you collect to put into the data buffer? Since in the former case this does need to be taken into account when comparing to other baselines in terms of sample efficiency.
- One suggestion in terms of presentation, Section 3 quite clearly explains the proposed approach, but since it does contain many different pieces, I believe it would be helpful to include some kind of pseudocode in either the main text or appendix to summarize Section 3.

---

> ### Author Response · Authors · 2022-08-02
> **Replies to Reviewer MUd3 (part 1/2)**
>
> > The paper argues why the additive edit function is preferred and the authors provide some solid observations for why this is the case. I was wondering if you could give a bit more intuition on why this is the case (especially why additive is better than overwrite).
>
> With the 'overwrite' design, the safety editor (SE) has to directly output a safe action $a$ instead of modifying the action proposal $\hat{a}$ by the utility maximizer (UM). Our assumption is that $\hat{a}$ is often reasonably safe already (imagine that the agent is far away from any obstacle, then a utility maximizing action should be somewhat safe). As a result, SE's output $a$ can be close to $\hat{a}$ in order to achieve a small loss (Eq. 7b). To introduce this inductive bias explicitly to SE, we can instead adopt the 'additive' design, where SE is only responsible for outputting the delta change $\Delta a$ on top of $\hat{a}$, without having to learn the (mostly) identity mapping. Across different states, this $\Delta a \approx 0$ bias makes the optimization landscape of SE easier. Please also see Figure 12 Appendix D for empirical results about 'additive' vs 'overwrite'.
>
> > Also a related question to the above, have you considered other forms of edit functions aside from additive and overwrite?
>
> We also considered a multiplicative edit function where SE outputs a scaling vector in $[-1,1]$ which is multiplied to the action proposal $\hat{a}$ in an element-wise way. However, this design can only 'shrink' and/or 'flip' action values. It is not as flexible as the current additive design where the SE can have total control of the final edited action in $[-1,1]$ by having $\hat{a}+2\Delta a$.
>
> Note that because SE can be arbitrarily complex, its output $\Delta a$ could depend on the current state $s$ and the action proposal $\hat{a}$ in an arbitrarily complex way. Thus even though the additive operation is simple, the overall editing process is already general enough to represent any modification.
>
> > When you use a batch of size $N$, you mentioned that you don't necessarily wait for the episode to finish when doing evaluation. Does this mean each batch could contain multiple episodes or a partial unfinished episode?
>
> We use parallel environments to collect data. A batch of _rollout_ samples contains $N$ independent environments, each with a rollout trajectory of length $L$ (a partial unfinished episode). Thus a batch could contain multiple partial unfinished episodes. The value of $L$ could be much smaller than the episode length $T$, for example, we collect and evaluate the violation rate of a batch every 5 time steps even though each episode might be 1000 steps.
>
> Note that this rollout batch is different from the replayed training batch which usually has a much larger batch size.

---

> ### Author Response · Authors · 2022-08-02
> **Replies to Reviewer MUd3 (part 2/2)**
>
> > In general, larger $N$ is preferred from an accuracy perspective but is problematic in terms of sample efficiency. From a theoretical perspective you could get away with a smaller $N$ if the underlying MDP has a small mixing time. However I wonder in practice how did you chose $N$?
>
> Correct, to reduce evaluation noise (especially when our safety threshold is very small) a larger rollout batch is preferred, which includes 1) more parallel environments, and 2) more rollout steps per evaluation.
>
> A larger rollout batch could decrease sample efficiency. For this reason, we compare all methods using the same rollout setup. We used 32 parallel envs with a rollout length of 5 for the SafetyGym tasks, and 16 parallel envs with a rollout length of 5 for the SafeCarRacing tasks. The absolute numbers don't matter much as long as 1) all compared methods have the same setup, and 2) they don't introduce too much noise in safety evaluation.
>
> Updated (Aug 07): We assume that the mixing time refers to the time the agent needs for reaching a steady state distribution from initial states. Our evaluation suffers less from the mixing time issue due to two reasons. 1) Both our Safety Gym and safe racing tasks have *randomized* environment layouts and agent initial positions, making the mixing time very small, and 2) we use multiple parallel environments for rollout and safety evaluation, alleviating the mixing time issue.
>
> > Also for evaluating $\Lambda$, did you have to collect a separate batch of data or was it the same data you collect to put into the data buffer? Since in the former case this does need to be taken into account when comparing to other baselines in terms of sample efficiency.
>
> It's the same data we put into the replay buffer. There is really no need to use the same rollout policy to collect another batch for evaluating $\Lambda$. Thus the sample efficiency comparison to other baselines is fair.
>
> > Section 3 quite clearly explains the proposed approach, but since it does contain many different pieces, I believe it would be helpful to include some kind of pseudocode in either the main text or appendix to summarize Section 3.
>
> Thanks for the suggestion, we'll add a pseudocode section in the appendix.
>
> > I wonder if you have considered or whether it's possible to integrate some of the ideas from Stooke et al. (2020) (referenced in your paper) which addresses the oscillation issue but for Lagrangian methods.
>
> The main contribution of this paper is to propose a two-policy framework. As a first-order approach, we basically directly adopted the classic Lagrangian method (with a slight modification of using rollout data instead of replayed data for safety evaluation), to balance utility and safety. As acknowledged in the limitations section, oscillation issue does exist as expected. Addressing this issue with ideas from [Stooke et al. (2020)] is kind of orthogonal to our focus here, but we believe it's quite possible to do so given that [Stooke et al. (2020)] basically improves the dynamics of the Lagrangian multiplier with PID control (Algorithm 2 in their paper), a procedure that can be plugged into our approach to replace the current integral control. Note [Stooke et al.2020] introduces two additional hyperparameters, thus it takes more efforts to tweak. Again, tweaking PID control is orthogonal to our focus.

---

### Author Response · Authors · 2022-08-02
**Rebuttal summary**

We thank the reviewers for their constructive comments. In summary, there are three major questions among reviewers and we highlight them here. For other questions/concerns, please see our detailed  replies to individual reviewers.

1.  *Why the two-policy design with an additive editing function and why it is better (reviewer MUd3, iJnb, and AoPc).*

	SEditor decomposes one policy that learns both utility and safety, into a utility maximizer (UM) and a safety editor (SE). UM only maximizes utility without resolving the conflict between utility and safety. SE edits UM's action proposal when it's unsafe. The learning tasks of UM and SE are two easier subtasks compared to learning one policy for the following reasons.

	a.  **Different effective horizons (SE and UM).** In most scenarios, safety requires either responsive actions ([Dalal et al 2018]), or planning a number of steps ahead to prevent the agent entering non-recoverable states. Although SE's behavior can in theory be very long-term (see our response to question 2 below), its actual decision horizon could be short depending on the nature of the safety. This is in contrast to UM's decision horizon which is usually long for goal achieving. In other words, we could expect the optimization problem of SE to be easier than that of UM. So SE has a chance of being learned faster if separated from UM.

	b.  **Constraint sparsity (SE).** Our inductive bias for SE is "to consider action safety only when necessary". The motivation is that usually constraint violation is only triggered for some states. And most often, the action proposal by UM is already safe if the agent is far away from obstacles. To explicitly introduce this bias, we use the additive editing function which ensures that the majority of SE's optimal outputs $\Delta a$ are close to 0. Across different states, this $\Delta a \approx 0$ bias makes the optimization landscape of SE easier. See Figure 12 for some empirical observations.

	c.  **Guarded exploration (UM).** From the perspective of UM, its MDP (precisely, action space) is altered by SE. UM's actions are guarded by the barriers set up by the SE. Instead of discouraging an unsafe action by UM (i.e., punishing the agent with negative signal), SE actually gives suggestions to UM by redirecting the unsafe action to a safe but also utility-high action to continue UM's exploration. This guarded exploration leads to a better overall exploration strategy because safety constraints are less likely to hinder UM's exploration (Figure 4 illustration).

	The above three advantages are our motivations for SEditor's two-policy design and the reasons why its better than the baselines. We will elaborate on them in the revised Section 3.

2.  *Whether the safety policy SE takes into account long-term safe behaviors, so that the agent won't enter non-recoverable states where by then SE can't do anything to keep it safe (Reviewer iJnb and AoPc).*

	Unlike the safety layer [Dalal et al 2018] which assumes projecting unsafe actions based on instantaneous safety costs, the training objective (Eq 7b) for SE uses a safety critic $Q_c$ which is learned as the expected future constraint return. SE policy that maximizes this critic will take into account long-term safety behaviors. In other words, if there are indeed non-recoverable states where SE can't do anything, SE will edit the agent's actions long before to avoid entering those states.

3.  *Three random seeds might be too few to be statistically conclusive on Safety Gym tasks (Reviewer hrVA).*

	We followed [Ray et al 2019] to use three random seeds. But we do agree that 3 seeds might not be statistically conclusive. So we've run 6 more seeds for safety gym tasks (now in total 3+6=9 seeds), for all methods.

	The updated training curves (Figure 3) is at  https://i.imgur.com/0d3YIoJ.png.

	The updated SWU scores (Table 1) is below. The overall results have almost no difference with before. SEditor's overall score is now higher (1.04 -> 1.10), in particular it becomes much better on PP2 with 9 seeds. Moreover, according to the definition, two SWU scores are best compared by division. Now we also show the improvement of SEditor over others in the table.

|    |CP1|CG1|CB1|CP2|CG2|CB2|PP1|PG1|PB1|PP2|PG2|PB2|SR|SRO|Overall|Improvement|
|------:|-----|-----|-----|-----|-----|-----|-----|-----|-----|-----|-----|-----|-----|-----|-----|----|
|  SAC|0.02|0.02|0.01|0.01|0.01|0.01|0.05|0.02|0.01|0.04|0.01|0.01|0.14|0.03|0.03|3567\%|
|PPO-Lag|0.01|0.11|0.00|0.00|0.04|0.01|0.00|0.13|0.03|0.00|0.09|0.01|0.04|0.04|0.04|2650\%|
|FOCOPS|0.03|0.28|0.03|0.03|0.20|0.05|0.02|0.32|0.10|0.01|0.26|0.09|0.04|0.04|0.11|900\%|
|SAC-actor2x-Lag|0.74|0.63|0.70|0.59|1.00|0.81|0.60|1.00|0.89|0.37|0.94|0.64|0.37|0.08|0.67|64\%|
|SEditor|1.01|0.94|0.78|1.49|0.99|0.95|1.55|1.00|1.00|1.78|1.00|1.00|1.28|0.57|**1.10**|-|

(The 2 safe racing tasks already use 9 seeds, so we didn't run extra seeds.)

---

### Author Response · Authors · 2022-08-05
**Updated the manuscript**

Dear reviewers, we've updated the manuscript (including appendix) to reflect the promised changes in our responses. The updates are highlighted in blue. The numbering of sections, figures, tables, and equations hasn't changed, so the references in existing reviews and responses are still valid.

---

### Meta-Review · Area_Chair_MnS2 · 2022-08-26

**Recommendation:** Accept
**Confidence:** Less certain

**Metareview:**

Reviews and responses make sense. The authors made a lot of improvements during the review process. The updated version could be an accepted.

**Award:**

No

---

### Decision · Program_Chairs · 2022-09-14

Accept